# A RNA producing DNA hydrogel as a platform for a high performance RNA interference system

Jaejung Song [1], Minhyuk Lee[2], Taeyoung Kim[2], Jeongkyeong Na[3], Yebin Jung[4], Gyoo Yeol Jung [1,3], Sungjee Kim[1,4] & Nokyoung Park[2]

RNA interference (RNAi) is a mechanism in which small interfering RNA (siRNA) silences a target gene. Herein, we describe a DNA hydrogel capable of producing siRNA and interfering with protein expression. This RNAi-exhibiting gel (termed I-gel for interfering gel) consists of a plasmid carrying the gene transcribing siRNA against the target mRNA as part of the gel scaffold. The RNAi efficiency of the I-gel has been confirmed by green fluorescent protein (GFP) expression assay and RNA production quantification. The plasmid stability in the I-gel results in an 8-times higher transcription efficiency than that of the free plasmid. We further applied the I-gel to live cells and confirmed its effect in interfering with the GFP expression. The I-gel shows higher RNAi effect than plasmids in free form or complexed with Lipofectamine. This nanoscale hydrogel, which is able to produce RNA in a cell, provides a platform technology for efficient RNAi system.

[1] School of Interdisciplinary Bioscience and Bioengineering, Pohang University of Science and Technology (POSTECH), 77 Cheongam Ro, Nam Gu 37673 Pohang, South Korea. [2] Department of Chemistry, Myongji University, 116 Myongji Ro, Yongin 17058 Gyeonggi-do, South Korea. [3] Department of Chemical Engineering, POSTECH, 77 Cheongam Ro, Nam Gu 37673 Pohang, South Korea. [4] Department of Chemistry, POSTECH, 77 Cheongam Ro, Nam Gu 37673 Pohang, South Korea. These authors contributed equally: Jaejung Song, Minhyuk Lee. Correspondence and requests for materials should be addressed to N.P. (email: pospnk@mju.ac.kr)

DNA has been mainly considered as a genetic material in biological systems, but there have been recent tremendous efforts to use these nucleic acids for various generic applications, such as building blocks in nanotechnology, carriers for drug delivery, and for highly efficient protein production[1–5]. Those efforts have mostly been based on the attractive chemical and physical properties of DNA molecules. For example, DNA can be synthesized into specifically designed sequences, and folded or assembled into very sophisticated structures with nanoscale accuracy. Moreover, these DNA nanostructures (e.g., nanorobots[6]) have revealed their own structural functions and have been used as templates for the precise assembly of inorganic materials[7]. In addition to their structural advantages, the biocompatibility and molecular recognition ability of DNA molecules and their assembled structures have driven DNA materials to wide biological applications. Among the materials fabricated by the assembly of DNA nanostructured building blocks, the DNA hydrogel (Dgel) has attracted considerable attention because of its various important applicabilities, such as in diagnostics[8,9], plasmonics[10], and drug delivery and release systems[11,12]. In addition, the Dgel has been employed for efficient protein production systems, being broader and more practical than other DNA nanostructures in terms of biological applications[13].

RNA interference (RNAi), a process in which gene expression is silenced using short RNA fragments, has been widely studied and has attracted huge general interest in the field of gene therapy[14–16]. The short RNA fragment (usually called short interfering RNA (siRNA)) interferes with and reduces gene expression by inducing the sequence-specific degradation and subsequent translation inhibition of complementary mRNA. Although many potential synthetic short RNA fragments that have effectively silenced the expression of a specific mRNA have been developed, there are still challenges in applying them for therapeutic purposes, mainly due to the biological instability of naked siRNA[17]. Most of the naked siRNAs are too unstable to be delivered into a target cell in an intact form, and the activity of the siRNA is insufficient to interfere effectively with the mRNA translation. To overcome this weakness of naked siRNAs and to realize the clinical application of this promising therapeutic method, the development of biocompatible molecular carrier systems that can efficiently deliver siRNAs in an active form to the target site is crucial[18,19].

Herein, we propose a conceptually new approach for RNAi by using the unique advantages of the Dgel, such as its high RNA productivity and improved stability compared with free plasmid DNA and RNA. The Dgel is fabricated by cross-linking short palindromic sticky ends of branched DNAs, via enzymatic reactions under physiological conditions, to form three-dimensional (3D) networked structures. One of the most important advantages of the Dgel is its ability to incorporate a particular gene into a DNA network by co-cross-linking of the branched DNAs. The system mainly comprises a gel matrix and a plasmid into which the gene fragment is inserted. More importantly, according to a previous report, the gene-incorporated Dgel could produce a gene-specific protein with an extremely high yield of up to 300 times compared with that from a linear plasmid under the expression conditions in cell lysates. In that report, it was shown that one of the main reasons for the high yield of protein production from the Dgel is its considerably high transcription efficiency, producing an ~50 times larger amount of mRNA than the free plasmid. The extraordinarily high RNA productivity of the Dgel and its gel structure inspired us to employ it as a high-performance RNA-producing machinery within a cell.

## Results

**RNAi-exhibiting gel (I-gel) development.** In this RNAi approach, we delivered nanoscale Dgels into a cell and allowed them to produce RNA, which can interfere with translation of the target mRNA and hence reduce its protein expression (Fig. 1a), instead of using the conventional method of directly delivering synthesized RNA fragments in free form or in complex with polymers. By delivering the Dgel, which can substitute for siRNA transfection, we expected improved stability of the delivery carrier and a higher interference efficiency resulting from its larger siRNA productivity in the cell compared with the intact form of siRNA. To validate our proposed RNAi concept, we designed a

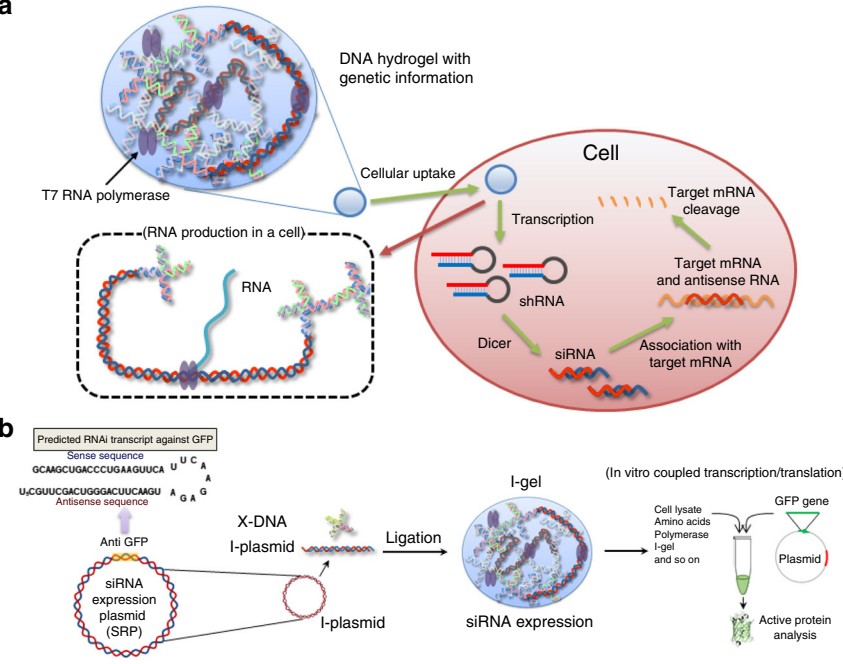

**Fig. 1** Illustration of RNAi mechanism of I-gel. **a** A schematic diagram illustrating the I-gel mechanism in a living cell. **b** A schematic diagram illustrating the I-gel mechanism in a cell lysate assay

green fluorescent protein (GFP)-interfering Dgel (denoted I-gel for interfering gel), and optimized the gel formation and components using cell lysates. Furthermore, we evaluated the interference efficiency of the I-gel within a mammalian cell line by comparing it with the free plasmid DNA that included the gene fragment, which could be transcribed to produce small RNA fragments that interfere with GFP expression (hereafter, denoted as I-plasmid)[20] and the I-plasmid complexed with Lipofectamine, which is the most popular commercial siRNA transfection reagent. Additionally, naked short hairpin RNAs (shRNAs) and shRNAs complexed with Lipofectamine were used as the controls for the cell experiments. The I-gel was synthesized by cross-linking the linearized plasmid DNAs with X-shaped DNAs (X-DNAs) (Fig. 1b). GFP was chosen as the target because it is a reporter protein, and the expression level of an active protein can be easily quantified through the simple measurement of fluorescence intensity[21]. The I-plasmid, which was linearized by a restriction enzyme and covalently incorporated into the I-gel matrix, included an shRNA-producing gene moiety.

**Evaluation and optimization of the I-gel**. To evaluate and optimize the I-gel, transcription and subsequent inhibition of GFP expression were performed using a coupled transcription and translation system (via a commercial cell lysate system), which provides a cell-mimic environment. This commercially available cell-free expression system made from a eukaryotic cell line (HeLa) is able to produce proteins from plasmids and can thus reveal the protein production ability of a plasmid or inhibition efficiency of a siRNA. Herein, to evaluate the interference efficiency of the I-gel, eight groups of cell lysates carrying different plasmid components were prepared as followings: no plasmid addition, as a blank control, GFP plasmid only, as a positive reference, GFP plasmid and scrambled shRNA, 1 eq (meaning that the copy number of shRNA added was the same as that of the I-plasmid), GFP plasmid and scrambled shRNA, 100 eq (meaning that the copy number of shRNA added was 100 times that of the I-plasmid), GFP plasmid and scrambled free plasmid mixture which could produce scrambled shRNA, GFP plasmid and Dgel composed of an X-DNA and scrambled shRNA-producible plasmid mixture, GFP plasmid and I-plasmid mixture, to measure the inhibition of free plasmid, and GFP plasmid and I-gel mixture, to evaluate the interference ability of the gel. In addition, we confirmed that there was no interference effect of the X-DNA and blank gel on GFP expression (Supplementary Figure 1). By measuring the fluorescence intensity of each expressed product and calculating the amount of active GFP in the solution, we could analyze the expression of GFP from each set of plasmid components and their associated RNAi efficiency. The blank control showed almost zero fluorescence intensity, verifying that there was no source of GFP expression in the lysate. In addition, the results from the controls with scrambled shRNA and a plasmid mixture expressing scrambled shRNA proved that there were no off-target effects of mismatched RNAs on the interference of GFP expression, except with the perfectly matched shRNA transcribed from free- or gel-phased I-plasmid (Fig. 2a, b). For the HeLa cell lysate group with only GFP plasmid added, a considerable amount of fluorescence intensity was observed, indicating that the plasmid had produced active protein in an appropriate way. In the case of cells transfected with the mixture of GFP plasmid and I-plasmid (57 ng), no significant change of fluorescence intensity was observed compared with that from the positive control (GFP plasmid only). This result means that the siRNA produced from the I-plasmid was not enough to interfere with the GFP expression in terms of amount or activity. For the mixture of the I-gel (containing 57 ng of I-plasmid) and

GFP plasmid, remarkably a significant decrease of fluorescence intensity was observed, where only less than 25% was measured compared with that from both the GFP plasmid only and the mixture of GFP plasmid and I-plasmid, indicating that the I-gel had effectively blocked expression of the protein. The fourfold difference in fluorescence intensity could clearly be recognized in an image taken by a digital camera (Fig. 2b inset). When we increased the added amount of free I-plasmid to the lysates, a significant reduction in the fluorescence intensity was observed, with 2000 ng of I-plasmid showing a similar interference effect as that of the I-gel containing 57 ng of I-plasmid (Supplementary Figure 2). From those results, it was confirmed that the I-gel was much more effective at interfering with the GFP expression. To maximize the RNAi efficiency of the I-gel, the gel fabrication process was optimized by conducting the GFP expression assay in the presence of various components of the I-gel under diverse experimental conditions. For every set of experiments, GFP plasmid-only expression was used as a positive control to minimize the error from cell lysate variation, and the same amount of GFP plasmid (1000 ng) was added for each reaction to compare the interference efficiency correctly. First, the amount of I-gel used in each reaction was varied from 0.1 to 2.0 μL for a total 25.0 μL reaction (including 1000 ng of GFP plasmid), where the corresponding amount of I-plasmid in each gel volume was 5.7, 11.4, 28.5, 57.0, and 114.0 ng, respectively. As a control, a series of reactions were also conducted using the same amount of I-plasmid and GFP plasmid in the cell lysate solutions. As shown in Fig. 2c, the I-gel exhibited higher RNAi efficiency than the I-plasmid under every gel amount. To further optimize the I-gel system, we varied the X-DNA and I-plasmid ratio from 1000:1 to 6000:1 by changing the X-DNA concentration of each gel while keeping the plasmid amount constant (Supplementary Figure 3). As seen in Fig. 2d, 1,500:1 (X-DNA:I-plasmid) was the optimum ratio for the efficient inhibition of GFP expression. With the other X-DNA:gene ratios, the decrease in GFP fluorescence intensity tended to be similar or slightly lower than 1500:1. Specifically, when 1.0 μL of I-gel (including ~57 ng of linearized I-plasmid) with the X-DNA:I-plasmid ratio of 1500:1 was added, the GFP production was only about 20% of that from the free I-plasmid. Consequently, the calculated RNAi efficiency of the gel was 57.6 times higher than that of the free I-plasmid under this condition (Fig. 2c, d).

**Quantification and comparison of the siRNA efficiencies**. To understand the reason for the high RNAi efficiency of the I-gel, we quantified the siRNA amount in each lysate after the expression reaction, and further investigated the mRNA level of the GFP plasmid transcribed in the presence of free I-plasmid and I-gel, respectively[22]. The comparison of RNA amount in each lysate solution was performed by the reverse-transcription quantitative polymerase chain reaction (RT-qPCR), where each RNA was reverse transcribed to complementary DNA (cDNA) by an enzymatic reaction with reverse transcriptase, and the relative amount of DNA was then measured by real-time monitoring of the fluorescence intensity of the PCR-amplified DNA. After reverse transcription of the RNAs extracted from the lysate solution, the resultant cDNAs were amplified to produce a sufficient amount for analysis. The amplified cDNA was run on an electrophoresis gel to confirm if it was the correct amplification product of the reverse-transcribed RNA (Fig. 3a). As can be seen in the image, only one gene product was amplified in each sample, verifying that the RT-PCR had produced only the intended cDNA. The different mobilities of the shRNAs from the I-plasmid and I-gel were due to the low specificity of the T7 RNA polymerase on the stop sequence. Although we had inserted a

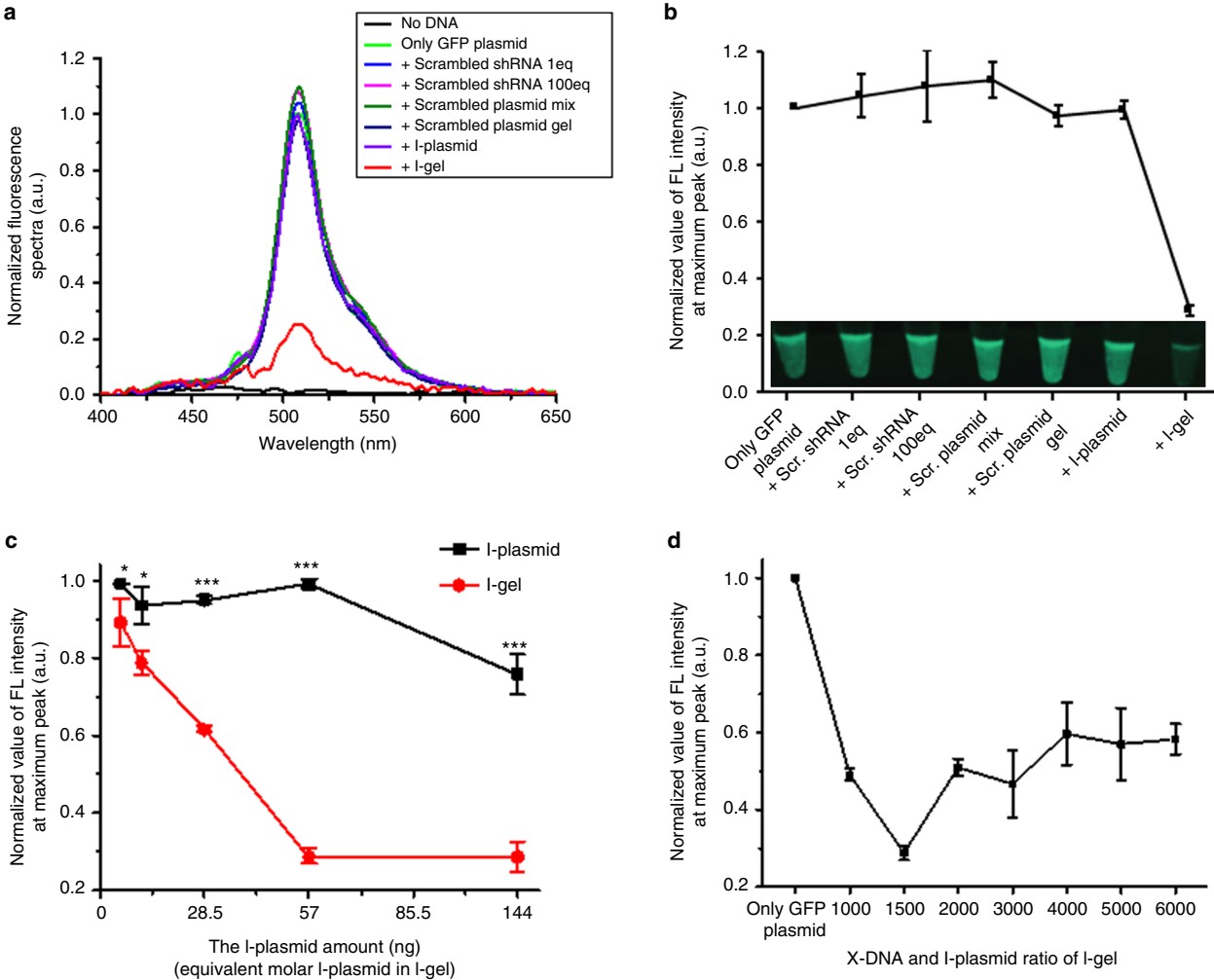

**Fig. 2** Optimization and evaluation of I-gel. **a** The FL spectra of GFP in each condition (no plasmid, only GFP expression, addition of scrambled shRNA 1 eq, scrambled shRNA 100 eq, scrambled plasmid mixture, scrambled plasmid gel, I-plasmid, and I-gel). **b** A comparison of the fluorescence intensities at the peak points of the graph (**a**) and the inset is fluorescence images of each sample. The relative FL intensity (**c**) with the addition of increasing amount of I-plasmid and I-gel (*$P < 0.05$, **$P < 0.005$ and ***$P < 0.001$, analyzed by $t$ test. Asterisks indicate statistically significant differences between I-plasmid and I-gel), (**d**) with the variation of X-DNA and I-plasmid ratio in I-gel. For all experiments, 57 ng of I-plasmid has been used except (**c**). This data was obtained by in vitro transcription/translation with HeLa cell lysate system. Error bars refer to standard deviations from three replicates

termination sequence at the end of the shRNA-transcribing region (Supplementary Figure 4), it would appear that the T7 polymerase did not to stop transcribing exactly on the sequence. According to the agarose gel electrophoresis image and melting temperature measurements of the reverse-transcribed DNAs (Fig. 3a, Supplementary Figure 5), the lengths of the shRNAs transcribed from the free I-plasmid and I-gel were 71 and 111 bases, respectively. Both were longer than the originally designed standard shRNA (56 bases). Since the completed PCR usually produces similar saturated amounts of DNA, showing a similar band intensity in the two lanes representing reverse-transcribed and amplified DNAs from the shRNAs produced by the free I-plasmid and I-gel, respectively, the relative amounts of siRNAs from the I-plasmid and I-gel were precisely measured by real-time PCR (qPCR) with appropriately designed primers. As expected, each sample showed the exponential, linear, and plateau phases of amplification as a function of cycle number (Supplementary Figure 6). The inset table in Supplementary Figure 6 compares the average threshold cycle (CT) values for the samples, which give information on the initial amount of cDNA in the sample before amplification started, where a higher CT value means a smaller amount of initial cDNA and vice versa[23]. In the

quantitation, we had calibrated the amount of RNA using standard samples (Supplementary Figure 6). The shRNA standard sample was a commercially synthesized one, whereas the mRNA standard was extracted from the transcribed solution. For a more accurate quantification, a spike-in control RNA (cel-miR-39) was employed for every RNA RT-qPCR. By adding the spike-in control to the expression lysate before extraction, the extraction rates for the RNA samples from the lysates could be calculated (Supplementary Table 1, 2). As shown in Fig. 3b, when 57 ng of I-plasmid was used, $18 \pm 3$ and $131 \pm 9$ copies of shRNA were produced from a single template of free I-plasmid and I-gel-incorporated I-plasmid, respectively. In addition to the comparison of the shRNA amount from the I-gel and free I-plasmid, the mRNA amount transcribed from the GFP plasmid in the presence of I-gel and free I-plasmid was evaluated to confirm that the reduced GFP fluorescence intensity is mainly due to the interfered mRNA level. The relative amount of GFP mRNA was measured using the same process as that for shRNA. As seen in the gel electrophoresis analysis, only one gene product was amplified (Fig. 3d), and it is clear that the amount of cDNA converted from the GFP-expressing lysate in the presence of I-gel was much lower than that from the GFP-expressing lysate with and without the

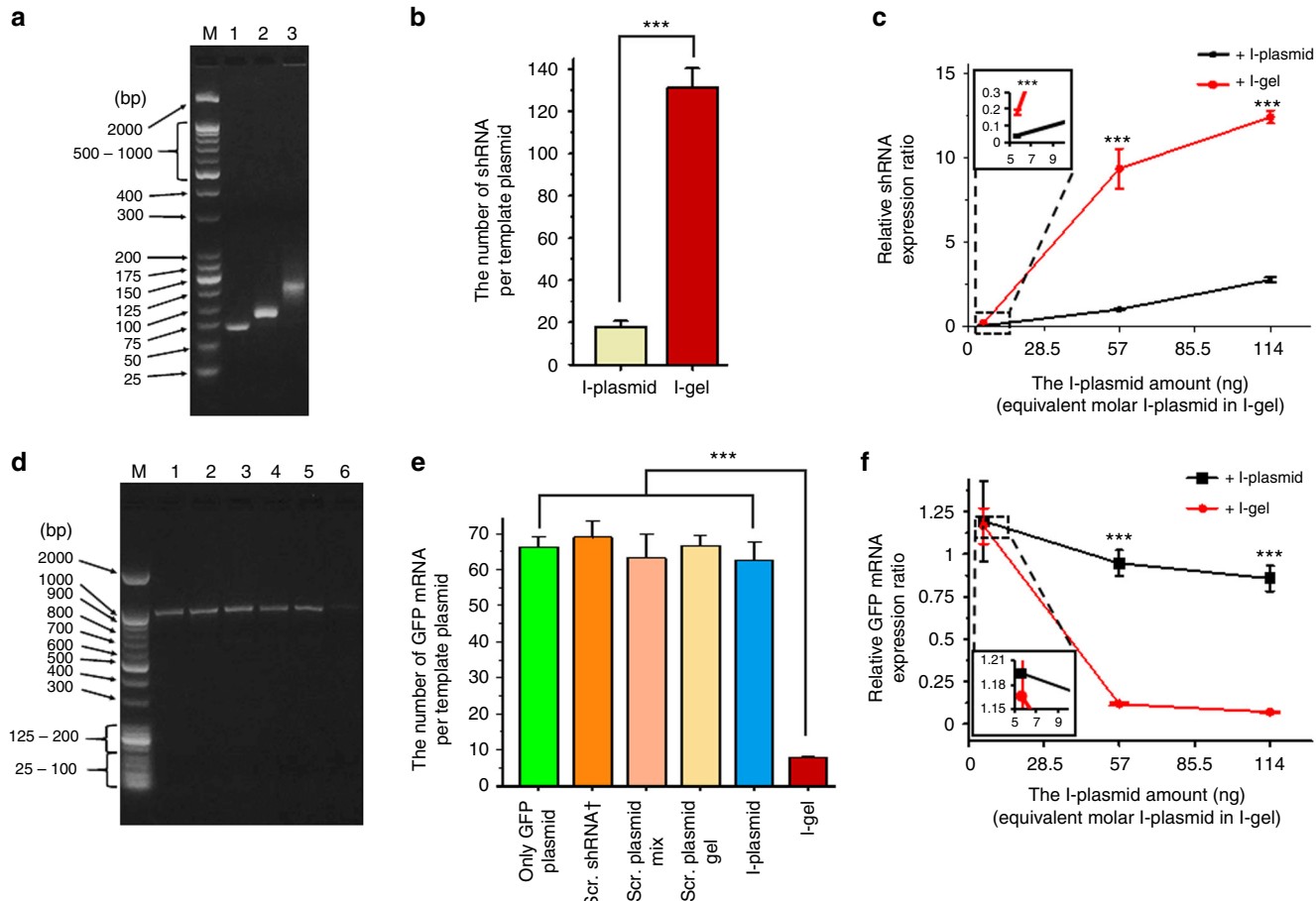

**Fig. 3** Comparison of RNA expressions. **a**, **d** The RT-PCR products in agarose gel electrophoresis for (**a**) shRNA and (**d**) GFP mRNA detection in extracted RNA samples from the cell lysates. **a** Samples on gel are lane M: molecular weight standard marker; cDNAs reverse transcribed respectively from Lane 1: standard shRNA, Lane 2: shRNA expressed from I-plasmid, Lane 3: shRNA expressed from I-gel; (**d**) cDNAs reverse transcribed respectively from the GFP mRNAs produced in the cell lysates transcribed/translated in the presence of Lane 1: only GFP plasmid, Lane 2, 3, 4, 5, 6: the addition of scrambled shRNA, scrambled plasmid mixture, scrambled plasmid gel, I-plasmid, and I-gel. **b**, **e** Copy number of RNA expressed from a template plasmid for (**b**) shRNA and (**e**) GFP mRNA expression in each condition (†concentration of scrambled shRNA was 102-fold increased in consideration of the template to RNA transcription rate of the I-gel) (**b**) (***$P < 0.001$, analyzed by $t$ test. Asterisks indicate statistically significant differences between I-plasmid and I-gel) (**e**) (***$P < 0.001$, analyzed by $t$ test. Asterisks indicate statistically significant differences between each condition marked by line and I-gel). **c**, **f** The relative RNA expression value for (**c**) shRNA and (**f**) GFP mRNA in 5.7 ng, 57 ng and 114 ng of I-plasmid amount condition. The insets of (**c**) and (**f**) represent zoomed up images of dashed box (***$P < 0.001$, analyzed by $t$ test. Asterisks indicate statistically significant differences between I-plasmid and I-gel). Error bars refer to standard deviations from three replicates

free I-plasmid. Under the same amount of GFP plasmid (1000 ng), $63 \pm 5$ and $8 \pm 1$ copies of GFP mRNA were produced from a single template GFP plasmid co-expressed with 57 ng of free I-plasmid and with I-gel containing 57 ng of I-plasmid, respectively (Fig. 3e). These data imply that the transcription efficiency of the I-gel is 7.3-folds that of the free I-plasmid, and the interference effect of the I-gel is around 7.9 times higher than that of the free I-plasmid at these optimized conditions. According to the additional experiments with different conditions of free I-plasmid and I-gel amounts in the cell lysates, the transcribed amounts of shRNA and mRNA (also GFP production in terms of fluorescence intensity) matched consistently with reverse correlation (Figs 2c, 3c, 3f). According to measurements of the amount of GFP produced in terms of fluorescence intensity, with 57 ng of I-plasmid, it was not easy to translate the shRNA production efficiency of the free I-plasmid and I-gel directly to interference efficiency, because the free I-plasmid showed little reduction of fluorescence whereas the I-gel showed a 70% reduction of intensity. When 114 ng of I-plasmid was used, however, the reduction of fluorescence intensity (interference effect) by the free

I-plasmid was 22.5%, which was around three times lower than that of the I-gel (70.1%) (Fig. 2c). The three times lower reduction rate of the free I-plasmid was attributed to its 4.5 times lower transcription rate compared with that of the I-gel (Fig. 3c). Although the fold changes of shRNA production and interference effects are not perfectly matched, the trend of an increased shRNA production rate with a higher interference effect seems consistent.

From this quantitative analysis of shRNA and mRNA amounts, we could conclude that the high transcription efficiency of the I-gel is a direct reason for the significant interference of GFP expression[24,25].

**Gene-silencing effect by the I-gel on a cellular level**. The successful demonstration of the high shRNA transcription rate and subsequent inhibition of GFP expression by the I-gel in cell lysates encouraged us to investigate the I-gel's RNAi effect in living cells. As previously emphasized, the use of the I-gel was expected to take advantage of the Dgel's own merits, such as its

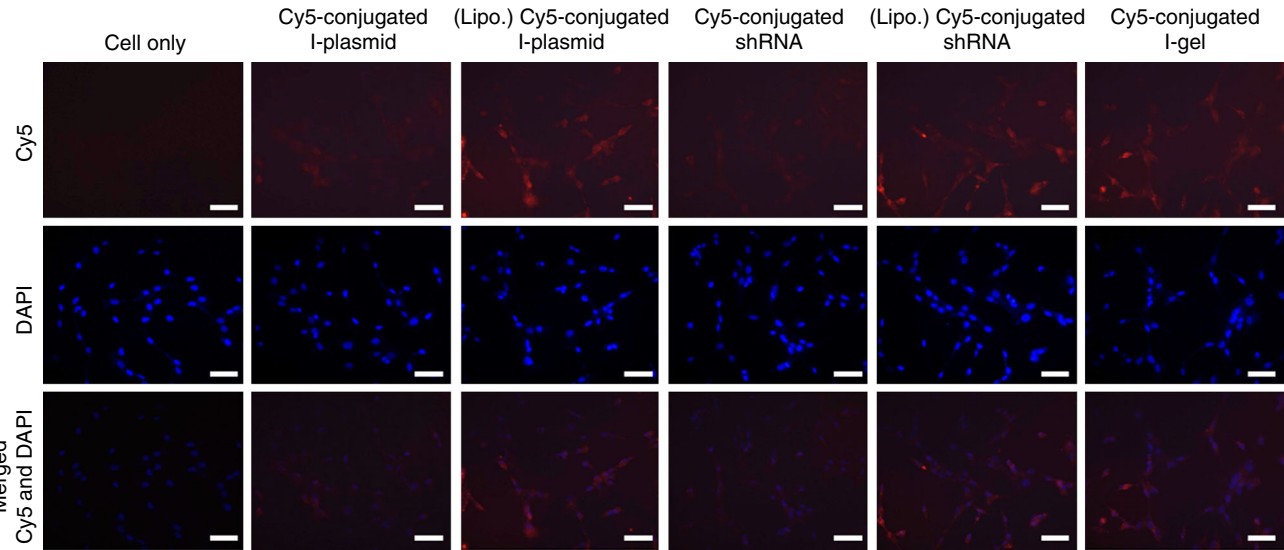

**Fig. 4** Comparison of intracellular uptake of plasmids in different formats. Second from left: I-plasmid labeled with Cy5, third: I-plasmid labeled with Cy5 complexed with lipofectamine, fourth: shRNA labeled with Cy5, fifth: shRNA labeled with Cy5 complexed with lipofectamine and the last: I-gel labeled with Cy5. Scale bar: 20 μm

high stability against enzymatic degradation, considerable RNA expression, and high cellular uptake and accumulation as benefitted from its size. Most mammalian cells, such as GFP-expressing Madin-Darby canine kidney (MDCK-GFP) cells, are known to internalize nanostructures and can engulf them through receptor-mediated endocytosis or membrane rupture upon nonspecific binding. According to the images taken after the incubation of MDCK-GFP cells with dye (Cy5)-conjugated I-gel in Dulbecco's modified Eagle's medium (DMEM; Hyclone, Logan, UT, USA) at 37 °C under 5% $CO_2$, a high accumulation of the I-gel was observed inside the cells (Fig. 4). In comparison with the cellular uptake efficiency of the other controls, the free I-plasmid showed little cellular uptake, whereas the I-plasmid complexed with Lipofectamine showed similar cellular uptake efficiency as the I-gel. In the case of shRNAs, the trend of cellular uptake of naked shRNAs and of those complexed with Lipofectamine was consistent with that of the I-plasmid (Fig. 4 and Supplementary Figure 7a). These results imply that the I-gel can be transported to and accumulated within the cell as efficiently as a form complexed with Lipofectamine.

Having confirmed that the I-gel could pass through the cell membrane and accumulate within the cell, we further studied if it can efficiently express shRNA with RNA polymerase and elicit the RNAi effect in live cells. To evaluate the RNAi efficiency of the I-gel more critically, the I-plasmid complexed with Lipofectamine (a well-known commercial transfection reagent) and the free I-plasmid were used as controls in addition to the blank control. Additionally, the cells were treated with other controls (viz. naked shRNA, Lipofectamine-complexed shRNA, scrambled shRNA, scrambled plasmid mixture, Lipofectamine-complexed scrambled plasmid, and Lipofectamine-complexed scrambled plasmid gel) and the GFP fluorescence intensities from the cells were compared. All cases, including I-gel, free I-plasmid, and Lipofectamine-complexed I-plasmid, were preincubated with T7 RNA polymerase for binding with the T7 promoter and then treated to MDCK-GFP cells in DMEM at 37 °C under 5% $CO_2$[26]. In the case of the controls, the same amount of I-plasmid (both in free and in Lipofectamine-complexed forms) as that in the I-gel was used for each interference experiment, except in the blank control where no plasmid was added. For the shRNA controls, 100-folds of the number of I-plasmid template was added to the

cells, considering the transcription efficiency of the plasmids in the I-gel. Each sample was observed after 48 h of incubation with the cells, and the relative GFP expression levels and correlating RNAi effects were quantified by fluorescence imaging and fluorescence-activated cell sorting (FACS) analysis. According to the fluorescence images (Fig. 5a, Supplementary Figure 8), a significant difference in the fluorescence intensities was observed between the I-gel and the other controls. In contrast to the other controls (including blank, free I-plasmid, naked shRNA, and all scrambled plasmids or siRNA), from which significant fluorescence intensities appeared, almost no fluorescence was observed with the I-gel. Only the Lipofectamine-complexed I-plasmid and Lipofectamine-complexed shRNA showed significantly reduced fluorescence intensity, which might be due to the high cellular uptake with the help of Lipofectamine. Since no cytotoxicity had been observed for the I-gel (Supplementary Figure 9), the reduced fluorescence can be considered as an interference result from the transcribed shRNA. The fluorescence images were further quantified on the basis of the average pixel intensity values, revealing the GFP expression interference effect of the I-gel to be 9.4-times and 2.8-times higher than that of the free I-plasmid and Lipofectamine-complexed I-plasmid, respectively (Supplementary Figure 7b). The fluorescence intensities of the cells were further analyzed using FACS (Fig. 5b). Compared with the blank control, in which the fluorescence intensity distribution in MDCK-GFP cells within a predetermined range was 59.6%, a considerable distribution shift to 10.2% had occurred for the I-gel-treated cells. In contrast, cells treated with either free I-plasmid or Lipofectamine-complexed I-plasmid showed relatively insignificant distribution shifts to 50.3% and 34.1%, respectively. For a more accurate and quantitative comparison of the interference effect of the I-gel on GFP expression, the GFP mRNA and shRNA in the cells were quantified using RT-qPCR (Fig. 5c, d). As was done in the lysate experiment, a spike-in control RNA (cel-miR-39) was employed for every RNA RT-qPCR. As shown in the results, the I-gel interfered with the transcription of GFP mRNA most effectively in the cells (only one fourth of the GFP mRNA amount was measured compared with that in non-treated cells). Only shRNA complexed with Lipofectamine showed a similar interference effect as the I-gel. In the case of shRNA, which had existed in the cells, the amounts transcribed from the

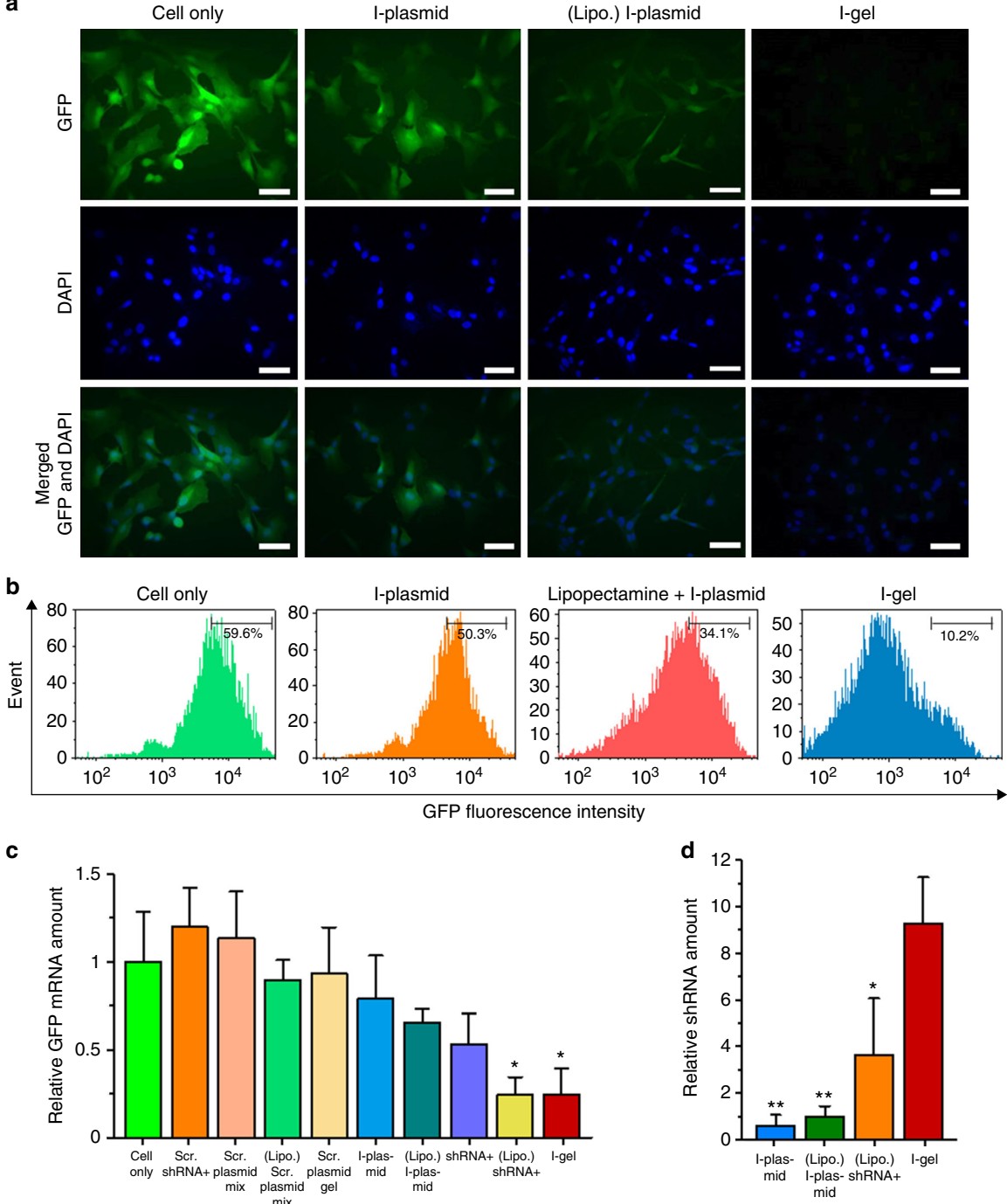

**Fig. 5** Gene-silencing effect by I-gel i n a cellular level. **a** Fluorescence images of MDCK-GFP-expressing (MDCK-GFP) cells coincubated with I-plasmid, I-plasmid complexed with lipofectamine, and I-gel. Scale bar: 20 μm. **b** Fluorescence-activated cell sorter analysis of GFP-expressing cell line, MDCK-GFP cells after treatment of sample/polymerase complexes in serum-deficient medium. The number means the percentage of GFP-overexpressing cells sorted within a prefixed gate region as indicated by a bar. **c**, **d** The relative amounts of (**c**) GFP mRNA and (**d**) shRNA from GFP-expressing MDCK cells after incubation in various conditions. (†concentration of these conditions were 102-fold increased in consideration of the template to RNA transcription rate of the I-gel) (**c**: *$P < 0.05$, analyzed by one-away ANOVA, followed by Holm–Bonferroni multiple comparisons post test. Asterisks indicate statistically significant differences between between each condition and Cell-only) (**d**: *$P < 0.05$ and **$P < 0.005$, analyzed by $t$ test. Asterisks indicate statistically significant differences between each condition and I-gel). Error bars refer to standard deviations from three (**c**) or six (**d**) replicates

I-gel was 9 times and 2.5 times higher, respectively, than that produced from Lipofectamine-complexed I-plasmid and the amount of remaining shRNA that had been added one time initially as a form complexed with Lipofectamine. To confirm the significance of the GFP mRNA inhibition efficiency difference between the controls and I-gel, we have performed one-way ANOVA and Holm–Bonferroni post test among the multiple groups (Supplementary Table 3). When the cell-only condition

was compared with other groups using Holm–Bonferroni post test, only the Lipo + shRNA ($P = 0.012$) and I-gel ($P = 0.011$) conditions showed statistically significant differences. In addition, when I-gel was compared with other groups, only Lipo + shRNA showed similar mRNA interference effect. These results imply that the interference effect of the I-gel is much more superior to that of both the free and complexed I-plasmids, even in a cell as well as in a lysate solution. Additionally, to confirm the distribution of data follows a normal distribution, the Shapiro–Wilk test was performed (Supplementary Figure 10).

## Discussion

The main rationale behind the plasmid design to produce shRNAs as siRNAs is as follows[27]: first, it is well known that short hairpin-structured double-stranded RNAs (dsRNAs) with 3′ overhangs are more advantageous for forming an RNA-induced silencing complex, resulting in a much higher RNAi efficiency than that formed by single-stranded RNA (ssRNA) transcripts. Second, it is easier and kinetically more favorable to anneal a single strand which contains both sense and antisense RNA sequences into a dsRNA than to hybridize two separate sense and antisense ssRNAs into the dsRNA. Third, in terms of RNA production, the transcription of one long RNA strand from a plasmid is also more favorable than transcribing two separately located short RNA strands. The size and functionality of the I-gel need to be controlled in order to facilitate its cellular uptake for efficient in-cell transcription. As reported previously, we had developed a method for nanoscale Dgel fabrication by enzymatic cross-linking of X-DNAs, each of which is composed of three phosphorylated sticky ends including palindromic sequences and one non-phosphorylated blunt end[10]. To fabricate the nanoscale I-gel, the mixture of X-DNAs and I-plasmids (containing sequences of sticky ends complementary to the X-DNA sticky ends) was cross-linked by T4 ligase. The fabricated I-gel and its ingredients required for I-gel formation were confirmed through electrophoresis (Supplementary Figure 11), and the size of the I-gel was evaluated using dynamic light scattering measurement (Supplementary Figure 12).

The remarkably high RNAi efficiency of the I-gel can be attributed to several properties which are unique to it, as follows: the plasmid stability in a gel network, the high transcription efficiency of the plasmid in the gel phase, an elongated promoter region, the gel size, and the higher local concentration of I-plasmids in the I-gel. The plasmid stability in the gel network was confirmed by comparing the denaturation resistance of free I-plasmid and I-gel in a DNase-containing solution and followed transcription efficiency measurement of I-gel and free I-plasmid after digestion reaction. As shown in Supplementary Figure 13 and Table 4, the I-plasmids in the solution were significantly denatured remaining only 40% of the free I-plasmid in 24 h of incubation losing about 63% of transcription efficiency. In contrast, ~95% of the I-gel had remained even after 24 h of incubation with DNase (measured using ImageJ software) keeping similar transcription efficiency with that of nondigested I-gel. In addition to the high stability of the I-gel, its transcription efficiency was also higher than that of the free I-plasmid (Supplementary Figure 14). According to the time-series comparison of transcription efficiency of the free I-plasmid and I-gel, shRNA production from the I-gel increased until 120 min and decreased thereafter, whereas the free I-plasmid produced the highest amount of shRNA at the initial period and then showed a gradual decrease in production with time. This transcription profile of the free I-plasmid and I-gel correlated well with the stability of the I-plasmid whether in a free or a gel format. The correlation between transcription and interference efficiency was also

confirmed by time-series monitoring of the GFP expression in a lysate solution at a fixed free I-plasmid and I-gel amount (57 ng of I-plasmid). The interference efficiency ratio of GFP expression between the I-gel and I-plasmid reached the maximum value at 2 h of reaction in the cell lysate and decreased with increasing time. This means that transcription by the I-gel was so fast that the reaction was completed within 2 h, whereas the I-plasmid was still transcribing after 6 h (Supplementary Figure 15, 16). For the cellular-level experiment, the stability of the I-gel was also a key parameter to improving its interference effect. As already reported, the network structure of the Dgel showed much improved stability against nuclease, and it retained the intactness of the I-gel for a longer time to transcribe more shRNA in the cell. We also monitored the interference effect of the I-plasmid in the presence of X-DNA, which is not able to hybridize with the plasmid. The results showed that the simple addition of X-DNA to the lysate solution had improved the interference effect dramatically (33.6-fold reduction of fluorescence intensity compared with that from the I-plasmid only; Supplementary Figure 17), indicating that the X-DNA protected the plasmid from nuclease attack[28]. Besides the improved stability and transcription efficiency of the I-gel, its promoter location may also play an important role in its efficiency. As we can see in the I-plasmid design (Supplementary Figure 4), the promoter is positioned near the cleavage site, which is cross-linked to X-DNA for forming the gel network. Considering the persistence length of double-stranded DNA (dsDNA), the promoter region can be elongated and easily exposed to RNA polymerase, thereby allowing easy access of the polymerase for fast transcription. The gel size also played an important role in the higher RNAi efficiency of the I-gel. An I-gel diameter of ~100 nm is an appropriate size for its efficient cellular uptake and accumulation inside the cell, which is also an important factor for enhancing the transcription and high RNAi effect. Finally, the higher local concentration of I-plasmid in the I-gel may be another factor in the improved RNAi efficiency of the I-gel. Since the plasmids which are incorporated in the I-gel are cross-linked to X-DNA and located in the gel, the local concentration of plasmids is elaborated more than 10-folds compared with that of the same amount of free I-plasmids suspended in a lysate solution or cytosol. The high plasmid concentration makes it easier for RNA polymerase to transport from one plasmid to another, resulting in the improved turnover rate of transcription. Such effect of a high local concentration of plasmid was also proven in a control expression experiment using a mixture of nanoscale Dgel containing sticky ends on its periphery and linear plasmids embedded with sticky ends complementary to the Dgel ends. Although there was no covalent cross-linking between the gel and plasmids, the hybridization induced the plasmids gathering on the gel periphery, resulting in a high local concentration and accordingly high RNAi efficiency (Supplementary Figure 17). The combination of the described factors is believed to be the key parameters behind the high RNAi efficiency of the I-gel, and also shows the unique advantage of employing the nanoscale Dgel for siRNA production in a cell.

In this paper, we have demonstrated a hydrogel type of plasmid —I-gel, made by simple cross-linking with branched DNAs— which can be a high-performance RNAi reagent. In this proposed concept, the I-gel was delivered into cells and showed a RNAi effect to silence GFP expression. Compared with previously reported DNA vehicles for siRNA delivery, where DNA was used mainly for the delivery of presynthesized siRNA[29], our I-gel is unique in terms of a delivered object and takes advantage of the transcription process as an amplification step. In the case of presynthesized siRNA delivery, only the amount of delivered siRNA has an interference effect. In our concept, however, the I-gel behaves as a factory which produces shRNA as well as a

vehicle. In addition, because the transcription carried out on the I-gel produced 131 copies of shRNA per template plasmid, there is an amplification effect of the shRNA amount compared with the direct delivery of presynthesized shRNA. However, thorough studies of the I-gel and comparisons with other siRNA delivery vehicles are still needed to improve its applicability. Nonetheless, since the shRNA-producing gel has superior stability, cellular uptake efficiency, and silencing effect over those of the free and Lipofectamine-complexed I-plasmids, we believe that the I-gel has huge potential to be used as a virus-free and toxicity-free alternative RNAi method to the direct delivery of siRNAs. We envision that the gel format of RNA production will not only be applied soon for in vivo RNAi, but will also be further developed for the production of diverse types of RNA, such as for mRNA in a cell.

## Methods

**Synthesis and characterization of the I-gel.** The four oligonucleotides of X-DNA (X01-X04 DNA strands) were commercially synthesized by Integrated DNA Technologies (Skokie, IL, USA). The X-DNA sequences (Supplementary Table 5) were designed, prepared, and characterized by following the same procedures as described in the literature with slight modifications[10,13]. Lyophilized DNA strands in 1 μmol scale was collected by centrifugation at 14,000 $g$ for 15 s at room temperature. Each DNA strand was resuspended to 1.0 mM concentration in 1x TE buffer (pH 8.0). Each tube was shaken and mixed using MULTI-THERM$^{TM}$ (Benchmark Scientific, NJ, USA) at 1500 rpm for ~3 h to completely dissolve the lyophilized oligonucleotides. In total, 0.05 μmol (50 μl) of each DNA strand was combined to a 1.5 ml tube, and nuclease-free water was added with the total volume of oligonucleotide mixture to 500 μl. 100 μl of oligonucleotide mixture was distributed into 0.5 ml tubes. The tubes were placed in a thermocycler (Bio-Rad, CA, USA). The four types of oligonucleotides (X01 ~04) were annealed with the following conditions: initial denaturation at 95 °C for 10 min; and 65 °C for 2 min, 60 °C for 5.5 min, decrease by 1 °C maintaining 1 min at that temperature, 20 °C for 30 s, and decrease to 4 °C for stabilization and storage of X-DNA. Then, for buffer exchange, the annealed X-DNA solution (500 μl) in a filter unit (3-kDa $Mw$ cutoff) for microcentrifugation was centrifuged for 6 h at 15,000 $g$ and 4 °C to reduce the solvent volume. The filter unit was transferred to fresh centrifuge tube. In total, 100 μl about 4 °C nuclease-free water was added to the filter unit. The filter unit was centrifuged for 4 h at 15,000 $g$ and 4 °C to rinse X-DNA of remaining salts. The cutoff filter module should be placed upside-down into the tube and centrifuged at 2000 $g$ for 3 min at 4 °C. The addition of nuclease-free water and centrifugation were performed two more times using the same centrifuge tube to completely remove residual salt of X-DNA. The final volume of X-DNA solution will be ~300 μl. The siRNA expression plasmid (I-plasmid) was purchased and maxi-prepared by Cosmo Genetech (Seoul, South Korea). To construct the I-gels, the I-plasmids were linearized using the restriction enzyme BamHI. The size of the siRNA gene was 66 bp with the spacer (or 498 bp with the T7 promoter) (see Supplementary Figure 4 for the I-plasmid map). The DNA amounts of the samples were determined from their UV/Vis absorption value using a BioSpectrometer (Eppendorf, Hamburg, Germany). The X-DNA and linear plasmids were firstly mixed at a predetermined molar ratio in the presence of T4 DNA ligase (Promega, Madison, WI, USA) to synthesize the Dgel. For an X-DNA:I-plasmid ratio of 1500:1, we used 10.5 μL of 75 μM X-DNA and 5.25 μL of 100 nM plasmid in a 30 μL reaction volume. For the Blank-gel experiment, the same amount of X-DNA material as in the I-gel was ligated with T4 DNA ligase. 10 μL of each sample was mixed with 2 μL of gel loading buffer and electrophoresed on a 0.7 or 2% agarose gel at 100 V for 60 min. All DNAs (X-DNA, I-plasmid, Blank-gel, and I-gel) were confirmed by agarose gel electrophoresis (Supplementary Figures 3, 11, 18). The synthesized Dgels were desalted twice using Amicon 3-kDa $M_w$ cutoff microtubular centrifugal filters. For scanning electron microscopy (SEM) imaging, the samples were freeze-dried in a FreeZone freeze dryer (Labconco, Kansas City, MO, USA) until all the water had been removed. The dried samples were cracked to reveal their fresh surfaces. After being sputter-coated with a 2~ 3-nm Pt layer for 100 s, the morphologies of these hydrogel surfaces were observed at ~50,000× magnification using an S-3500N (Hitachi, Tokyo, Japan) scanning electron microscope at an accelerating voltage of 15 kV (Supplementary Figure 19).

**I-gel stability test.** Seven samples were prepared for each of the free I-plasmid and I-gel. In each sample, 2 μL of the free I-plasmid (57 ng) or I-gel (1:1500 gel containing 57 ng of I-plasmid) was diluted in 12 μL of distilled water then 4 μL of transcription optimized 5 × buffer was added and treated with $3.33 \times 10^{-5}$ units of DNase I (No. M6101; Promega) to 20 μL of final volume, followed by incubation at 37 °C for 0, 1, 4, 12, 24, and 48 h, respectively. After incubation, the 20 μL samples were separated into two 10 μL aliquots, one for electrophoresis and another for following transcription. In each of the aliquot, the denaturation reaction was terminated by the addition of 1 μL of RQ1 DNase stop solution and incubation for

10 min at 65 °C. Thereafter, 10 μL of each sample was mixed with 2 μL of gel loading buffer and electrophoresed on a 2% agarose gel at 100 V for 60 min (Supplementary Figure 13). Another aliquot of each sample was used for transcription reaction to demonstrate the transcription efficiency of I-gel after digestion reaction. shRNAs were transcribed from the I-gel or free I-plasmid templates (0, 24, 48 h) using transcription kit (Riboprobe; Promega) following the manufacturer's instruction. The obtained shRNAs were quantified by RT-qPCR as described in Total RNA preparation and reverse transcription and Real-time PCR and data analysis sections (Supplementary Table 4).

**Expression and inhibition analyses of proteins.** Coupled transcription and translation kits (1-Step Human Coupled IVT Kit - DNA, Catalog no. 88882) were purchased from Thermo Fisher Scientific (Waltham, MA, USA), and the reactions were carried out by following the procedures suggested by the manufacturer. In brief, the HeLa cell lysate, accessory proteins, reaction mix, and pcDNA3.1( + ) IRES GFP plasmid (0.5 μg/μL; No. 51406; Addgene, Cambridge, MA, USA) were mixed in a 12.5:2.5:5:2 (v/v) ratio and incubated at 30 °C for 1, 2, 4, and 6 h. After the indicated reaction times, the sample solutions were incubated for 24 h at 4 °C to both stop the reaction and ensure sufficient protein folding time. To evaluate the interference efficiency, free I-plasmid or I-gel was added to the cell lysate in the presence of 1000 ng of GFP plasmid. In the control experiments for the optimized I-gel condition containing 57 ng of I-plasmid (17.5 nmol), scrambled shRNA (17.5 nmol and 1.75 μmol; 1 eq and 100 eq to the amount of I-plasmid, respectively) (SN-1003; Bioneer, Seoul, Korea), scrambled plasmid mixture (17.5 nmol; scrambled shRNA expression plasmid mixture; Cosmo Genetech), scrambled plasmid gel (scrambled plasmid mixture enzymatically cross-linked with X-DNA), the I-gel component (X-DNA only, I-plasmid only, or these mixture without enzymatic cross-linking; that is, no gel formation), or Blank-gel (enzymatic cross-linking of X-DNAs only) were added directly to the GFP expression reaction solution. All reactions were performed at least three times. Unless otherwise stated, the reaction volume was kept at 25 μL. The fluorescence spectra were analyzed using a spectrofluorophotometer (FluoroMate FS-2; SCINCO Co., Ltd., Seoul, Korea) for determining the level of GFP expression in the cell lysate solution.

**Total RNA preparation and reverse transcription.** As a spike-in control, 3 μL of cel-miR-39 (33 fmol/μL; No. 59000; Norgen Biotek Corp., Thorold, ON, Canada) was added to all of the preexpressed cell lysates (20 μL) before RNA extraction. Then, total RNA was extracted from 23 μL of the lysate using TRIzol Reagent (Ambion, Thermo Fisher Scientific) according to the manufacturer's instructions. The extracted total RNA was dissolved in 10 μL of DEPC-treated water (Ambion, Thermo Fisher Scientific). Then, 2 μL of total RNA was polyadenylated with 1 mM ATP in 1× poly(A) polymerase buffer containing 5 U of $Escherichia\ coli$ poly(A) polymerase (New England Biolabs, Ipswich, MA, USA) at 37 °C for 30 min in a 20 μL reaction mixture. For reverse transcription, 4 μL of the tailed RNA was mixed with 1 pmol of RT primer and 0.5 mM dNTP mix and then made up to a volume of 13 μL with DEPC-treated water. The mixture was heated at 65 °C for 5 min and then quickly chilled on ice. The reaction mixture was then made up to a final volume of 20 μL by the addition of 5 mM dithiothreitol, 1 × first-strand buffer, 40 U RNase inhibitor, and 200 U SuperScript III Reverse Transcriptase (Invitrogen, Carlsbad, CA, USA) and incubated at 50 °C for 1 h, followed by enzyme inactivation at 70 °C for 15 min. The RT primer sequences (see Supplementary Table 5) were designed for the synthesis of cDNA from the RNA sequence. The primer used was the 3′ RACE adapter, which is provided in the FirstChoice RLM-RACE kit (Ambion, Thermo Fisher Scientific). All primers were synthesized by Cosmo Genetech, except for the cel-miR-39 forward primer (Norgen Biotek Corp.). The recovery rate of RNA in each lysate solution was estimated from the difference between the CT value of the obtained spike-in control and that of the reference cel-miR-39 (not proceeded with the extraction process).

**Real-time PCR and data analysis.** The amounts of shRNA and GFP mRNA were quantified by qPCR using the StepOne Real-Time PCR System (Applied Biosystems, Foster City, CA, USA). The qPCR of each sample was carried out in a 20 μL total volume with 2 μL of cDNA, 10 μL of 2 × Maxima SYBR Green qPCR Master Mix (Thermo Fisher Scientific), and 10 pmol of each primer. Amplification was performed with the following conditions: initial denaturation at 95 °C for 10 min; and 30 cycles of 95 °C for 15 s, 60 °C for 30 s, and 72 °C for 30 s. The qPCR was also performed with cel-miR-39 as a reference to obtain the recovery rate of total RNA in each sample. The qPCR protocol was the same as that described above in this section, except for the primers used, which were 10 pmol of a common reverse primer (same as the shRNA reverse primer) and cel-miR-39 forward primer. The melting curve of each amplified DNA product was obtained by measuring the fluorescence intensity change of the SYBR Green dye for six times per sample while increasing the temperature from 60 °C to 95 °C at a rate of + 0.3 °C/s after completion of the qPCR. The specific primers for qPCR were designed using the Primer3 program (http://primer3.ut.ee/) (Supplementary Table 5). The relative expression of the targets was determined by using the comparative $2^{-\Delta\Delta CT}$ method. The relative quantity of GFP mRNA was also confirmed through band intensity measurement on the 2% agarose gel. Amplification of the cDNA was performed

with the same PCR conditions and program as used for the qPCR described above, except for the PCR cycle number (20 for GFP mRNA).

**Calibration and quantification of the absolute RNA amount.** Standard curves for the RNA concentration and CT value were obtained by using each RNA standard material. For the shRNA, the standard curve was obtained by using synthesized shRNAs purchased from Integrated DNA Technologies. For the GFP mRNA, the curve was obtained using the GFP mRNA extracted from a transcription kit (Riboprobe; Promega) according to the manufacturer's instructions. The RNA amount of the standard shRNA was informed by Integrated DNA Technologies, whereas that of the extracted standard GFP mRNA was determined from its UV/Vis absorption value using the BioSpectrometer. The obtained standard RNAs were converted to cDNA by reverse transcription, as described above in the "Total RNA preparation and reverse transcription" section. After that, the cDNA was diluted 10, 100, 1000, and 10,000 times, and then qPCR was repeated three times, and the obtained CT values were used to obtain a standard curve. The CT values obtained from the RT-qPCR were converted to the initial RNA concentrations in the cell lysates through a calibration process (Supplementary Figure 6). The final absolute amount of RNA was determined by multiplying each RNA recovery rate (%) and the value of RNA amount from the calibration curve (Supplementary Tables 1, 2).

**Cellular labeling with Cy5-I-gel.** Madin-Darby Canine Kidney (MDCK) cells were obtained from the Korea Cell Line Bank (Seoul, Korea) and maintained in DMEM supplemented 10% fetal bovine serum and 1% penicillin–streptomycin in a moistened and $CO_2$ (5%)-maintained incubator. The MDCK expressing GFP (MDCK-GFP) cells were prepared by transfecting pEGFP-N1 vector into the cells with the use of Lipofectamine reagents (Thermo Fisher Scientific) according to the manufacturer's protocol. Afterward, we sorted the GFP-emitting cells using FACSCanto II system (BD Biosciences, Franklin Lakes, NJ, USA). All through the research, the cells were tested negative for mycoplasma contamination. Cultured MDCK-GFP cells in 24-well plates with coverslips (50,000 cells per each well) were coincubated with the Cy5-I-gel (Cy5-conjugated I-gel) containing Cy5-conjugated X01 DNA sequence strands (corresponding to 2.625 μM X-DNA; Integrated DNA Technologies) in serum-free medium for 4 h. To prepare the Cy5-I-plasmid, Cy5-dsDNA (Cy5-conjugated dsDNA) was firstly prepared by annealing Cy5-conjugated X01 (with an additional 5′-p-GATC sticky end) and its complementary counter-strand (5′-GAG GTA GGT ACG GAT CTG ACC GCT ATT CAT CGG TCG-3′). Then, the Cy5-dsDNA and I-plasmid were mixed and ligated at a molar ratio of 5000:1. Excess unbound Cy5-dsDNA was removed by centrifugation (12,400 g, 4 °C, 60 min) using Amicon microtubular centrifugal filters (50 kDa Mw cutoff) for 4–5 times. The Cy5-shRNA (Cy5-conjugated shRNA) was commercially synthesized (Integrated DNA Technologies). For the Cy5-I-plasmid and Cy5-shRNA sets, cells were coincubated with 2.625 nM of each sample in serum-free medium for 4 h. For the Lipofectamine-Cy5-I-plasmid and Lipofectamine-Cy5-shRNA sets, Lipofectamine and the Cy5-conjugated sample were electrostatically complexed by mixing them together in a molar ratio of 1:1 to a final solution concentration of 2.625 μM Lipofectamine and 2.625 μM substrate. Then, cells were coincubated with the 2.625 μM Lipofectamine-Cy5-I-plasmid or Lipofectamine-Cy5-shRNA in serum-free medium for 4 h, respectively. In the case of the cell-only control, the cells were coincubated with serum-free DMEM containing 1% (v/v) penicillin (HyClone). After 4 h of coincubation, all the samples were washed twice with phosphate-buffered saline (PBS) buffer (0.1 M, pH 7.4) and the cells were further incubated for 6 h in a growth medium containing 10% (v/v) fetal bovine serum (FBS; HyClone) and 1% penicillin to ensure sufficient cellular recovery and uptake time. The cultured cells were fixed with 4% formaldehyde for 10 min at room temperature and further washed three times with PBS buffer (0.1 M, pH 7.4). For microscope imaging, coverslips with the attached cells were mounted onto glass slides using aqueous mounting medium with an antifading agent. The fluorescence images were recorded using a Zeiss Axioplan 2 microscope, and pictures were taken using a Zeiss Axiocam HR camera.

**GFP gene-silencing experiment using the I-gel.** First, the I-gel containing 262.5 μM X-DNA and 175 nM I-plasmid was incubated with 300 U of T7 RNA polymerase (Thermo Fisher Scientific) for 15 min at room temperature. After the incubation, the I-gel sample was diluted to 100 times in serum-free DMEM containing 1% penicillin. Then, 1/6 times the volume of I-gel/polymerase complex was added to each well of a 24-well plate with coverslip which had been preplated with MDCK-GFP cells (20,000 cells per well) for 24 h. For the I-plasmid, scrambled plasmid mixture, and scrambled plasmid gel sets, MDCK-GFP cells were co-cultured with the same molar amount of each plasmid and T7 RNA polymerase as in the case of the I-gel. For measuring the RNAi effect of naked shRNA and Lipofectamine-shRNA, 100-fold more of each RNA sample was used relative to the number of I-plasmid template, taking into consideration the shRNA expression rate of the I-gel as calculated from the cell lysate experiment. For the naked shRNA and scrambled shRNA sets, MDCK-GFP cells were coincubated with 175 nM of the RNA sample. The Lipofectamine-complexed samples were prepared according to the manufacturer's instructions. For the Lipofectamine-I-plasmid and Lipofectamine-scrambled plasmid mixture sets, Lipofectamine and each plasmid

sample were electrostatically complexed by mixing them together in a molar ratio of 5:1 to obtain a final solution concentration of 8.75 nM Lipofectamine and 1.75 nM plasmid substrate. For the Lipofectamine-shRNA set, Lipofectamine and free shRNA were electrostatically complexed by mixing them in a molar ratio of 5:1 to obtain a final solution concentration of 875 nM Lipofectamine and 175 nM shRNA. The MDCK-GFP cells were coincubated with the as-prepared Lipofectamine-complexed samples in serum-free medium for 4 h. For the cell-only set, only MDCK-GFP cells were incubated in the serum-free medium for 4 h, without any samples. All the samples were washed out after the 4 h coincubation period, and the cells were additionally incubated for 48 h with the growth medium (DMEM containing 10% (v/s) FBS and 1% penicillin) to evaluate the RNA-silencing effect at 37 °C under 5% $CO_2$. The cells were then fixed with 4% formaldehyde for 10 min at room temperature and washed with PBS buffer (0.1 M, pH 7.4). For microscope imaging, the coverslip-attached cells were mounted onto glass slides using aqueous mounting medium with an antifading agent.

**Gene expression analysis by quantitative PCR in live cells.** All samples were treated to and incubated in each cell medium at the same amount and same condition as used in the previously described GFP gene-silencing experiment using the I-gel section. After incubation, the cell medium was aspirated from the cell culture plates, and the MDCK-GFP cells were washed once with PBS buffer and trypsinized to detach the cells in each well of the plate. The detached cells were diluted with PBS buffer to deactivate the trypsin, and were then collected into a 1.5 mL microtube and pelleted by centrifugation (180 g, 5 min). The supernatant was removed and the cells were diluted with 20 μL of PBS buffer. The cell suspension solution (20 μL) was treated with 1 mL of TRIzol Reagent, 3 μL of cel-miR-39 (33 fmol/μL), and 200 μL of chloroform solution. The subsequent RNA extraction was performed according to the manufacturer's instructions. For quantifying the relative amount of GFP mRNA and shRNA at the cellular level, all qPCRs of the sample preparations are carried out in the same method as described in the "Total RNA preparation and reverse transcription" and "Real-time PCR and data analysis" sections.

**Cell imaging analysis for pixel intensity quantification.** By processing of the cell image, data were conducted using the MATLAB (The MatWorks, Inc., Beltsville, MD, USA) program. The raw fluorescence images were imported to MATLAB, and each pixel in the images was converted to each component of the matrix. The distribution of the fluorescence intensities of each component was graphically represented by a histogram. The entire dataset was divided into 256 intervals.

**Fluorescence-activated cell sorting analysis.** The cells were washed once with 1 mL of PBS followed by aspirating the growth medium from the cell culture plates. Then, the cells were detached from the plate by trypsin treatment. The detached cells were collected into a 15-mL tube followed by centrifugation resulting a pellet (180 g, 3 min). The supernatant trypsin solution was removed and the cell pellet was washed twice using 2 mL of PBS. The cells were resuspended in PBS to be a concentration of 50,000 cells/mL. Then, the samples were prepared by fixing the cells in 1% formaldehyde solution for 10 min at room temperature. The samples were analyzed for GFP fluorescence intensity using FACSCanto II system (BD Biosciences, Franklin Lakes, NJ, USA).

**Relative viability levels in dark condition.** An MDCK-GFP cell suspension (5000 cells per well) was dispensed into a 96-well plate (Corning Inc., Corning, NY, USA) and incubated for 1 day at 37 °C under 5% $CO_2$. Once the cells had attached to the plate, they were coincubated with different concentrations of the I-gel (corresponding to the X-DNA concentration) in the growth medium (DMEM containing 10% (v/v) FBS and 1% penicillin) in the dark. The I-gel was composed of a X-DNA: I-plasmid molar ratio of 1500:1. These samples were incubated for 6, 24, and 48 h at 37 °C under 5% $CO_2$. At the end of each incubation time, Cell Counting Kit-8 solution (Dojindo Molecular Technologies, Kumamoto, Japan) was added to the samples according to the manufacturer's instructions. After further incubation for 1 h, the absorbance at 450 nm was measured using a microplate reader. The results of this measurement were expressed as the ratio between the absorbance of the sample and that of the negative-control cells without sample incubation.

**Statistical analysis.** Statistical analysis was performed with Prism 7.05 software (GraphPad Software) for Student's $t$ test, one-way ANOVA with Bonferroni multiple comparisons post test. The Holm–Bonferroni multiple comparisons post test was performed using Bonferroni multiple comparisons post test results from Prism 7.05 software (GraphPad Software)[30]. The normality test was calculated by the Shapiro–Wilk test. In results of the Shapiro–Wilk test, Data were approximately normally distributed. Statistical significance is indicated as $^*P < 0.05$, $^{**}P < 0.01$, $^{***}P < 0.001$.

## Data availability
The authors declare that data supporting the findings of this study are available within the article and its Supplementary Information files. All relevant data can be provided by the corresponding author upon reasonable request.

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

## Acknowledgements

This study was supported by the National Research Foundation of Korea (NRF) grant funded by the Korea government (MSIP) (NRF-2015R1A2A2A04005445). M.L. was supported partially by the Korea Institute of Energy Technology Evaluation and Planning (KETEP), and the Ministry of Trade, Industry & Energy(MOTIE) of the Republic of Korea (No. 20174010201160).

## Author contributions

N.P. conceived the project. N.P. and J.S. designed and developed the synthetic procedure for I-gel. J.S., M.L., J.N., and T.K. carried out the experiments. N.P., M.L., S.K., G.Y.J., J.S., J.N., and Y.J. analyzed the data. N.P., S.K., and J.S. wrote the paper. J.N., M.L., T.K., and Y.J. assisted with writing the "Methods" section in the paper. All authors discussed the results and commented on the paper.

## Additional information

**Competing interests:** The authors declare no competing interests.

