## [Peer Review File · Nature Communications]

Reviewers' comments:

Reviewer #1 (Remarks to the Author):

The paper describes the use of a DNA hydrogel that is injected into cells where it is transcribed to produce interfering RNA (siRNA) molecules that shut down the production of a protein. While the concept is interesting, the technical quality of the study is by far not sufficient to warrant publication (see details below). Even if one would believe the data that suggest suppression of GFP expression, the question would remain if this concept is applicable in a more general way, i.e., other proteins and other cells.

Fig. 2: The entire experiment is only based on the fluorescence of GFP expression – what if not iRNA but rather something else is responsible for the lack of protein expression? Hence, all experiments need a thorough quantification by rt-PCR, i.e., how many copies of siRNA and mRNA are produced per molecule template under each condition?

Fig. 2a: Why is there no inhibitory effect of the I-plasmid in the IVT experiment?

Fig. 2c: 114 ng of I-plasmid lead to 20% reduction of GFP expression. This result contradicts the data of Fig.2a where no decrease was detected.

Fig. 3a: Why has the siRNA obtained from I-gel (lane 2) a different mobility than the product obtained from plasmid (lane 1)? The proposed >50 fold increase in siRNA production is not supported by this gel. The data in Fig. 3 are from a single data point only. They are not sufficient to support the data of Fig. 2 (see above).

Fig. 3b, c, e, f: The data is not analyzed sufficiently. How many template molecules were present in the samples and how many were detected by RT-PCR (recovery rate)?

The text states that 8 times more siRNA was produced from the gel than from the plasmid. How does this translate to the findings of fluorescence analysis in Fig.2 ?

Fig. 4. The data would need to be supported by RT-PCR data that prove that indeed changes in siRNA and mRNA concentrations occur due to the addition of plasmids and I-gel.

Supplementary Fig. 1: inadequate description. What are the numerous samples indicated in 1a?

Supplementary Fig. 2: inadequate data. The existence of X-DNA is not supported. There is no obvious difference in lanes 3-9 that would support the increasing amount of X-DNA; there is no significant difference between a) and b) that would support the formation of a gel.

Supplementary Fig. 3: inadequate data. No bands visible in the gels that would support the formation of a gel. In other words: you cannot produce a gel containing only negative controls. Each condition requires at least one positive control also to warrant that the RT-PCR worked correctly.

Supplementary Fig. 5: unclear data. What should be shown with these two gels? What is the difference between bulk- or nano-scale I-gel? Apparently it does not make a difference when 4 (gel in a) or 3 (gel in b) sticky-ends X-DNA are used. What exactly is meant with sticky-ends X-DNA?

Supplementary Fig. 6: DLS indicates about 100 nm particle size – how does this agree with the SEM images (Fig. S12) where structures of hundreds of micrometers are shown?

The bibliography is by far not optimal. Please cite recent reviews from major players in the field

where appropriate; avoid old ones, such as refs 2, 5, 6, 7. Ref 8 does not concern DNA hydrogels.

Reviewer #2 (Remarks to the Author):

This article presents an interesting new way to induce gene silencing within cells using crosslinked DNA nanoparticles that have an encoded shRNA hairpin sequence within them. While the technology is novel for the gene silencing field and is interesting, there are concerns with controls for some of these studies, as well as other suggested revisions, as detailed in the comments below.

- There should be a control of the gel with a scrambled shRNA-GFP sequence. In RNA gene silencing work, scrambled sequences are important to verify that there are no off-target effects being observed.
- The low knockdown with lipofectamine + plasmid is surprising. Did the authors work on optimizing the lipofectamine experimental conditions?
- The results shown in figure 3 (qPCR) require a control. Did the authors run a standard for quantification of their samples, or an internal housekeeping gene as a control? The data cannot be normalized in this way (panels c and f) without appropriate controls / quantification.
- The entire article needs grammatical corrections.
- The authors should cite other works that have looked at siRNA delivery using DNA vehicles. For example there is a new article from Ren, et al., Nature Communications: <https://www.nature.com/articles/ncomms13580> . Also, what about other RNA-DNA particles that have been studied?

How does your work compare / contrast from other technologies?

- Have you measured the gel stiffness, inter-plasmid distance, compactness of dna network (p. 7 line 145) of these gels? If so these general relationships would be important to mention in this paper as well, particularly if it is believed that they impact the siRNA production. If not, obtaining some information with respect to these could be valuable for this work.
- Page 6, line 125: refrain from using the phrase 'so huge'. Replace with more scientific word/ phrase.
- Page 7, line 147: what do you mean by 57 ng of "genes"? Is that the weight of the linearized i-plasmid estimated to be in there? Should clarify.
- It is not apparent how the cytotoxicity studies were run, specifically how the I-gel and cells were co-incubated. More details are needed here for repeatability.
- The media that the cells were cultured in is not clear. Was it always serum-free DMEM or was there ever serum, were any antibiotics ever used, etc.?
- There is no mention of statistical analysis in the report.

Reviewer #3 (Remarks to the Author):

In the current study, I-gel is fabricated by crosslinking linearized plasmid DNAs (I-plasmid) with X-shaped DNAs (X-DNAs) to produce small RNA from the plasmid DNA and silence target gene expression. It is shown that I-gel has significant gene silencing effect on GFP expression compared to I-plasmid in the cell lysate system, although no rationale is given. In addition, control groups such as GFP-siRNA or GFP-shRNA should be employed to compare the gene silencing effect. Furthermore, additional experiments evaluating stability, intracellular uptake efficiency, and shRNA release of the I-gel would strengthen the manuscript to understand the merits of the I-gel.

1. According to Figure 2, I-gel significantly decreased GFP fluorescence intensity while I-plasmid did not affect at all. The authors should clearly describe why they show a difference in the interfering efficiency. The transcription efficiency of I-plasmid should be similar to that of I-gel (I-plasmid + X-DNA) in the presence of the RNA polymerase in the cell lysate assay because siRNA is

produced from I-plasmid in the I-gel. Thus, gene knockdown efficiency would be expected to be similar to each other. Does X-DNA in I-gel play special role in transcription?

2. How stable the T7 RNA polymerase would be in the I-gel and how long it is active in the I-gel. Because stability and activity of the polymerase would largely influence the transcription efficacy in this system, the authors should provide supporting results.

3. The authors need to explain how shRNA can be released from the I-gel and show the release profile. ShRNA can be physically captured in the I-gel. ShRNA release amount would affect subsequent gene silencing efficiency.

4. It would be better if qPCR result of in vitro cell lysate system (Figure 3) is combined with Figure 2. In particular, Ct curves (Figure 3b) can move to supplementary information.

5. On page 9, line 197, if the authors want to discuss the intracellular uptake mechanism of the I-gel, they should show the evidence. In the current manuscript, not only the mechanism, but also stability and RNA production of the I-gel were only based on the authors' speculation.

6. In Figure 4a, if the authors would like to highlight I-gel as an efficient gene carrier in terms of intracellular uptake, a positive control such as lipofectamine is required for comparison.

7. In Figure 4b, GFP-siRNA/lipofectamine is more suitable as a control instead of I-plasmid/lipofectamine to compare the gene silencing effect because I-gel contains RNA polymerase unlike I-plasmid/lipofectamine. That means I-plasmid/lipofectamine can produce siRNA only if it is transported into nucleus while I-gel can generate siRNA in the cytosol. In addition, neither free siRNA or shRNA is employed as a control in the current study.

8. qPCR should be performed to quantify the actual gene knockdown efficiency in living cells. Additional experiments are required.

Overall, this work falls short of the standards one would expect for a journal like NC. The writing of the manuscript is also sub-standard.

Apr. 18, 2018

Response to Reviewers:

Manuscript ID: NCOMMS-17-24306

Title: A RNA producing DNA hydrogel as a platform for a high performance RNA interference system

Dear Editor,

Thank you so much for the correspondence with the referees. We appreciate all the sincere comments by reviewers on our manuscript. The comments were carefully considered during the revision of the manuscript. In the upcoming pages, we tried to respond to these comments point by point. Our detailed list of changes and responses to reviewers' concerns are given below.

FYI: the blue letters indicate the revised part or added description in our revised manuscript.

Response to reviewer 1

Comments 1:

Fig. 2: The entire experiment is only based on the fluorescence of GFP expression – what if not iRNA but rather something else is responsible for the lack of protein expression? Hence, all experiments need a thorough quantification by rt-PCR, i.e., how many copies of siRNA and mRNA are produced per molecule template under each condition?

Our response: We thank the reviewer for the helpful comment. To confirm that the shRNA is responsible for the reduced GFP expression, we have performed additional experiments using several negative controls 1) scrambled shRNA mixtures, 2) plasmids mixture which can produce scrambled shRNA, and 3) DNA hydrogels which are composed of X-DNA and scrambled shRNA producible plasmid mixture. As shown in Figures R1a, b, direct addition of scrambled shRNA mixture has not shown any reduction of the GFP expression. In case of DNA hydrogels as well as plasmids, no interference effect has been observed either. Those results prove that the reduction of GFP expression is caused by the shRNA which has been transcribed from the I-gel and I-plasmid.

As the reviewer suggested, we have also performed RT-qPCR (reverse transcription quantitative polymerase chain reaction) to quantify the RNA amount in every cell lysate. In the quantitation, we have calibrated the amount of RNA using standard samples. The shRNA standard sample has been commercially synthesized and the mRNA standard has been extracted from transcribed solution. For more accurate quantification, a spike-in control RNA (cel-miR-39) has been employed for every shRNA and mRNA RT-qPCR. By adding the spike-in control to the expression lysate before extraction, extraction rates for RNA samples from the lysate could be calculated (Table R1 and R2). As shown in Figure R2b, 18 ± 3 and 131 ± 9 copies of shRNA have been produced from a single free I-plasmid template and an I-plasmid template in the I-gel, respectively. Under same amount of GFP plasmid (1000 ng), 63 ± 5 and 8 ± 1 copies of GFP mRNA have been produced (Figure R2e) when it is co-expressed with 57 ng of free I-plasmid and I-gel including the same amount of I-

plasmid, respectively. These data imply that the transcription efficiency of I-gel is 7.3 folds of free I-plasmid and the interference effect of I-gel is around 7.9 times higher than that of free I-plasmid.

We have added the results obtained from additional experiments and revised the manuscript accordingly.

In our revised manuscript:

pp5-6

Here, to evaluate the interfering efficiency of the I-gel, four separated sets of expression have been performed by adding different plasmid components to the cell lysate; 1) no additional plasmid as a blank control, 2) GFP plasmid only as a positive reference, 3) GFP plasmid and scrambled shRNA 1 eq (means the copy number of added shRNA is the same as that of I-plasmid), 4) GFP plasmid and scrambled shRNA 100 eq (means the copy number of added shRNA is 100 times of I-plasmid), 5) GFP plasmid and scrambled free plasmid mixture which can produce scrambled shRNA, 6) GFP plasmid and DNA hydrogel which are composed of X-DNA and scrambled shRNA producible plasmid mixture, 7) GFP plasmid and I-plasmid mixture to measure the inhibition of free plasmid, and 8) GFP plasmid and I-gel to see the interfering ability of the gel.

p9

In the quantitation, we have calibrated the amount of RNA using standard samples (Supplementary Fig. S7). The shRNA standard sample has been commercially synthesized and the mRNA standard has been extracted from transcribed solution. For more accurate quantification, a spike-in control RNA (cel-miR-39) has been employed for every RNA RT-qPCR. By adding the spike-in control to the expression lysate before extraction, extraction rates for RNA samples from the lysate could be calculated (Supplementary Table S1 and Table S2). As shown in Figure 3b, when 57 ng of I-plasmid has been used, 18 ± 3 and 131 ± 9 copies of shRNA have been produced from a single template of free I-plasmid and I-plasmid in the I-gel, respectively.

Comments 2:

Fig. 2a: Why is there no inhibitory effect of the I-plasmid in the IVT experiment?

Our response: We thank the reviewer for the helpful comment. In Fig. 2a in originally submitted manuscript, the I-plasmid amount used for the experiment was 57 ng which is the same with that used for I-gel. According to the results, there was little inhibitory effect with 57 ng of I-plasmid. As shown in Figure R3, when we increased the amount of the added I-plasmid, however, the inhibitory effect has been observed and 2,000 ng of I-plasmid has shown similar interference effect with that of I-gel including 57 ng of I-plasmid. This is due to the different transcription efficiency of I-plasmid and I-gel which has been shown in Figure R4.

We have added the additional results and revised the manuscript accordingly.

In our revised manuscript:

pp6-7

In case of introducing the mixture of GFP plasmid and I-plasmid (57 ng), no significant change of fluorescence intensity has been observed compared with that from the positive control, GFP plasmid only. This result means that the siRNA produced from I-plasmid is not enough to interfere the GFP expression in terms of amount or activity. As the last set of the comparison, the mixture of the I-gel where the same amount of I-plasmid as the above expression is incorporated within the gel matrix and GFP plasmid has been added to the cell lysate and the fluorescence intensity has been monitored after same period of time of the reaction. Remarkably, in this time, a significant decrease of fluorescence intensity has been monitored. Only less than 25% of GFP fluorescence intensity has been measured compared with that from both GFP plasmid only and mixture of GFP plasmid and I-plasmid expression indicating that the I-gel effectively blocked the GFP expression. **The four times of fluorescence intensity difference** could be clearly recognized in an image taken by usual digital camera (Fig. 2b inset). **When we have increased the added amounts of free I-plasmid to the lysates, significant amount of fluorescence intensity reduction has been observed and 2,000 ng of I-plasmid has shown similar interfering effect with that of I-gel with 57 ng of I-plasmid (Supplementary Fig. S2).**

Comments 3:

Fig. 2c: 114 ng of I-plasmid lead to 20% reduction of GFP expression. This result contradicts the data of Fig.2a where no decrease was detected.

Our response: The reviewer's concern is well taken. The fluorescence spectrum presented in Fig.2a in original manuscript has been obtained from a sample including 57 ng of I-plasmid. We are sorry for causing the confusion and we have revised the manuscript for clear understanding.

In our revised manuscript:

p6

In case of introducing the mixture of GFP plasmid and I-plasmid (57 ng), no significant change of fluorescence intensity has been observed compared with that from the positive control, GFP plasmid only.

Comments 4:

Fig. 3a: Why has the siRNA obtained from I-gel (lane 2) a different mobility than the product obtained from plasmid (lane 1)? The proposed >50 fold increase in siRNA production is not supported by this gel. The data in Fig. 3 are from a single data point only. They are not sufficient to support the data of Fig. 2 (see above).

Our response: We thank the reviewer for the helpful comments. The different mobility of shRNAs from I-plasmid and I-gel is due to the low specificity of the T7 RNA polymerase's termination sequence recognition. Although we have inserted a termination sequence at the end of shRNA transcribing region (Figure R5), the T7 polymerase seems not stop transcribing exactly on the termination sequence. According to the agarose gel electrophoresis image and melting temperature measurements of reverse transcribed DNAs (Figure R2a, Figure R6), the length of shRNA transcribed from free I-plasmid and I-gel are 71 bases and 111 bases, respectively. Both are longer than

standard shRNA (56 bases).

The increase of shRNA production of I-gel compared with that of I-plasmid is about 8 times according to our RT-qPCR results (Figure R2b). The 50 fold increase is not for our case but the case of previously reported protein producing gel. The increase may be different for each experimental condition, optimization and etc. The gel in Figure 3a in original manuscript has been obtained from the DNA samples resulting from the RT-PCR. Since the completed PCR usually produces similar saturated amount of DNA, the two lanes representing DNA reverse transcribed and amplified from shRNA produced from I-plasmid and I-gel have shown similar band intensity.

As we agree with the reviewer's comment on the insufficient data of Figure 3 to support the data of Figure 2 in our original manuscript, we have further performed additional experiments to obtain shRNA and GFP mRNA amounts produced under three different free I-plasmid and I-gel amount conditions in cell lysates. According to our new results, the transcribed amounts of shRNA and mRNA (also GFP production in terms of fluorescence intensity) have been consistently matched with reverse correlation (see Figure R7 which combines amounts of shRNA, mRNA, and GFP produced at three different I-plasmid amounts in forms of free and I-gel).

We have added the additional results and revised the manuscript accordingly.

In our revised manuscript:

pp8-9

The different mobility of shRNAs from I-plasmid and I-gel is due to the low specificity of the T7 RNA polymerase on the stop sequence. Although we have inserted a termination sequence at the end of shRNA transcribing region (Supplementary Fig. S5), the T7 polymerase seems not to stop transcribing exactly on the sequence. According to the agarose gel electrophoresis image and melting temperature measurements of reverse transcribed DNAs (Supplementary Fig. S6), the length of shRNA transcribed from free I-plasmid and I-gel are 71 bases and 111 bases, respectively. Both are longer than originally designed standard shRNA (56 bases). Since the completed PCR usually produces similar saturated amounts of DNA showing the similar band intensity for two lanes representing DNAs reverse transcribed and amplified from shRNA produced from free I-plasmid and I-gel, the relative amount of siRNA from I-plasmid and I-gel were precisely measured by qPCR with appropriately designed primers.

p10

Under same amount of GFP plasmid (1,000 ng), 63 ± 5 and 8 ± 1 copies of GFP mRNA have been produced from a single template GFP plasmid when it is co-expressed with 57 ng of free I-plasmid and I-gel including the same amount of I-plasmid, respectively (Fig. 3e). These data imply that the transcription efficiency of I-gel is 7.3 folds of free I-plasmid and the interference effect of I-gel is around 7.9 times higher than that of free I-plasmid at this optimized conditions. According to the additional experiments with several different conditions of free I-plasmid and I-gel amounts in cell lysates, the transcribed amounts of shRNA and mRNA (also GFP production in terms of

fluorescence intensity) have been consistently matched with reverse correlation (Fig. 2c, Fig. 3c and 3f).

Comments 5:

Fig. 3b, c, e, f: The data is not analyzed sufficiently. How many template molecules were present in the samples and how many were detected by RT-PCR (recovery rate)?

Our response: We thank the reviewer for the helpful comment. For more sufficient analysis, a spike-in control RNA (cel-miR-39) has been employed for every shRNA and mRNA RT-qPCR. By adding the spike-in control to the expression lysate, recovery rates for RNA samples from the lysate could be estimated. As shown in Table R2, the recovery rates of shRNA have been $36.2 \pm 2.0\%$ and $32.2 \pm 0.5\%$ from the lysate solutions expressed with free I-plasmid and I-gel, respectively. In the case of GFP mRNA (Table R1), $86.8 \pm 2.6\%$, $89.3 \pm 3.4\%$, and $59.0 \pm 1.7\%$ of mRNAs have been recovered from the lysates expressed without interference, with free I-plasmid, and with I-gel, respectively.

We have added the additional results and revised the manuscript accordingly.

In our revised manuscript:

p9

In the quantitation, we have calibrated the amount of RNA using standard samples (Supplementary Fig. S7). The shRNA standard sample has been commercially synthesized and the mRNA standard has been extracted from transcribed solution. For more accurate quantification, a spike-in control RNA (cel-miR-39) has been employed for every RNA RT-qPCR. By adding the spike-in control to the expression lysate before extraction, extraction rates for RNA samples from the lysate could be calculated (Supplementary Table S1 and Table S2). As shown in Figure 3b, when 57 ng of I-plasmid has been used, 18 ± 3 and 131 ± 9 copies of shRNA have been produced from a single template of free I-plasmid and I-plasmid in the I-gel, respectively.

Comments 6:

The text states that 8 times more siRNA was produced from the gel than from the plasmid. How does this translate to the findings of fluorescence analysis in Fig.2 ?

Our response: The reviewer's concern is well taken. All the data points and pictures of Figure 2 of our original manuscript were obtained at the 57 ng of I-plasmid and I-gel. The 8 times of shRNA has been obtained from comparison of the shRNA amounts produced respectively from 57 ng I-plasmid and the same amount of I-plasmid incorporated I-gel. Each shRNA might interfere the GFP expression in the lysate resulting different amounts of produced GFP which have been translated into different fluorescence intensities as shown in Figure 2. According to the fluorescence intensity measurements, with 57 ng of I-plasmid, it is not easy to directly translate the shRNA production efficiency of free I-plasmid and I-gel because free I-plasmid has shown little reduction of fluorescence while 70% of intensity has been reduced under I-gel. When 114 ng of I-plasmid has been used, however, the reduction of fluorescence intensity (interference effect) of free I-plasmid is 22.5% which is around 3

times lower than that of I-gel, 70.1%. The three times lower reduction rate of free I-plasmid has been attributed to the 4.5 times lower transcription rate (see Figure R2f) of free I-plasmid compared with I-gel. Although the fold numbers of shRNA productions and interference effects are not perfectly matched, the trend of increased shRNA production rate on the higher interference effect seems consistent.

We have added an appropriate discussion on our revised manuscript.

In our revised manuscript:

p10

According to measurements of the amount of produced GFP in terms of fluorescence intensity, with 57 ng of I-plasmid, it is not easy to translate the shRNA production efficiency of free I-plasmid and I-gel directly to interference efficiency because free I-plasmid has shown little reduction of fluorescence while 70% of intensity has been reduced under I-gel. When 114 ng of I-plasmid has been used, however, the reduction of fluorescence intensity (interference effect) of free I-plasmid is 22.5% which is around 3 times lower than that of I-gel, 70.1%. The three times lower reduction rate of free I-plasmid has been attributed to the 4.5 times lower transcription rate of free I-plasmid compared with I-gel. Although the fold numbers of shRNA productions and interference effects are not perfectly matched, the trend of increased shRNA production rate on the higher interference effect seems consistent.

Comments 7:

Fig. 4. The data would need to be supported by RT-PCR data that prove that indeed changes in siRNA and mRNA concentrations occur due to the addition of plasmids and I-gel.

Our response: We thank the reviewer for the helpful comments. Following the reviewer's suggestion, we have thoroughly studied the shRNA and mRNA amount level in live cells using RT-qPCR. According to the results, GFP mRNA level in I-gel treated cell has shown about 25% of that transcribed in non-treated cell (cell only). Only shRNA complexed with Lipofectamine treated cell has shown similar interference effect with I-gel treated one (Figure R8d). For the case of shRNA, the relative amounts of shRNA in the cell have been measured using RT-qPCR and the result have been summarized in Figure R8e)

We have added the additional results and revised the manuscript accordingly.

In our revised manuscript:

P13

For a more strict and quantitative comparison of the interference effect of I-gel against the GFP expression, the GFP mRNA and shRNA in the cells have been quantified using RT-qPCR (Fig. 4d and 4e). As it has been done in lysate experiment, a spike-in control RNA (cel-miR-39) has been employed for every RNA RT-qPCR. As shown in the results, I-gel has interfered the GFP mRNA transcription most effectively in the cells (only one fourth of the GFP mRNA amount was measured compared with non-treated cells). Only shRNA complexed with

lipofectamine has shown similar interference effect with the I-gel. In case of shRNA, which has existed in the cells, the shRNA amounts transcribed from the I-gel was 8 times and 2 times higher than that produced from lipofectamine-I-plasmid and remained shRNA that had been added as a complexed form with lipofectamine, respectively.

Comments 8:

Supplementary Fig. 1: inadequate description. What are the numerous samples indicated in 1a?

Our response: We thank the reviewer for the helpful comment and sorry for causing the confusion. The Supplementary Figure 1 shows that there is no interference effect of both X-DNA which has been used as a gel matrix and the gel with no I-plasmid incorporated within the gel (BL-gel indicating blank gel). To make it clear we have presented all fluorescence spectrum obtained from triplicated experiments for each X-DNA and blank gel. But since the many spectrums might induce confusion as the reviewer pointed out, we have removed spectrums except a representative one for each sample. We have revised the Figure accordingly.

Comments 9:

Supplementary Fig. 2: inadequate data. The existence of X-DNA is not supported. There is no obvious difference in lanes 3-9 that would support the increasing amount of X-DNA; there is no significant difference between a) and b) that would support the formation of a gel.

Our response: We thank the reviewer for the helpful comment. We have re-performed the agarose gel electrophoresis, and replaced the Supplementary Figure 2 in our original manuscript with the newly obtained data (Figure R9). In the new agarose gel image, the DNA intensity in each well are clearly increasing accordingly with the higher amount of X-DNA used for I-gel formation. The two agarose gel images are supposed to be similar because both are DNA hydrogel samples. The only difference is that the right samples are large size of DNA hydrogels and the left ones are nanoscale DNA hydrogels. Since the DNA hydrogels, regardless of large or small sizes, are crosslinked and produced into large molecular DNA structures by ligation of X-DNAs and I-plasmids, both are expected to remain in the wells. However, we have removed the agarose gel electrophoresis image of bulk (large) size DNA hydrogels from Supplementary Figure 2 since it still may cause a confusion.

Comments 10:

Supplementary Fig. 3: inadequate data. No bands visible in the gels that would support the formation of a gel. In other words: you cannot produce a gel containing only negative controls. Each condition requires at least one positive control also to warrant that the RT-PCR worked correctly.

Our response: We thank the reviewer for the helpful comment. We re-obtained gel electrophoresis images including positive controls and replaced the Supplementary Figure 3 with the new one (Figure R10).

Comments 11:

Supplementary Fig. 5: unclear data. What should be shown with these two gels? What is the difference between bulk- or nano-scale I-gel? Apparently it does not make a difference when 4 (gel in a) or 3 (gel in b) sticky-ends X-DNA are used. What exactly is meant with sticky-ends X-DNA?

Our response: We thank the reviewer for the helpful comment. Supplementary Figure 5 is presented to show that 3 sticky-ends X-DNA as well as 4 sticky-ends X-DNA can make hydrogels by enzymatic crosslinking.

Bulk-scale I-gel means a large size DNA hydrogel which is synthesized in a mold by crosslinking 4 sticky-ends X-DNAs and I-plasmids. When the 4 sticky-ends X-DNAs and I-plasmids are crosslinked in a mold with an appropriate concentration, the gel is formed into the similar size and shape of the mold. We call this large mold shape gel as a bulk-scale I-gel. On the other hand, if we crosslink the I-plasmids with 3 sticky-ends X-DNAs, nano-sized hydrogel was synthesized of which size can be measured using DLS and can be seen under SEM. We call this small gel as a nano-scale I-gel. For the agarose gel images, as previously mentioned, since the DNA hydrogels, regardless of large or small sizes, are crosslinked and produced into large molecular DNA structures by ligation of X-DNAs and I-plasmids, both are expected to remain in the wells.

X-DNA means a X-shaped DNA nanostructure with 4 branches which are fabricated by hybridization of 4 partially complementary oligonucleotides. With an appropriate designing the sequence of each oligonucleotide, each end of branches of X-DNA can be introduced with sticky end or blunt end. If three branches have sticky ends and one branch of a X-DNA has blunt end, we call this X-DNA as a 3 sticky-ends X-DNA. And if all four branches have sticky ends, we call this as a 4 sticky-ends X-DNA.

To eliminate the possibility of causing confusion, we have removed the data about bulk DNA hydrogel from the Supplementary Figure 5 and revised the manuscript accordingly (see Supplementary Fig. S12).

Comments 12:

Supplementary Fig. 6: DLS indicates about 100 nm particle size – how does this agree with the SEM images (Fig. S12) where structures of hundreds of micrometers are shown?

Our response: The reviewer's concern is well taken. The SEM image of our original manuscript has been taken from a "bulk-scale I-gel" to present porous structure of I-gel. However, as the reviewer mentioned, the scale is not agreed with the "nano-scale I-gel", we have added the SEM image taken from nano-scale I-gel in our revised manuscript (Figure R12).

Comments 13:

The bibliography is by far not optimal. Please cite recent reviews from major players in the field where appropriate; avoid old ones, such as refs 2, 5, 6, 7. Ref 8 does not concern DNA hydrogels.

Our response: We thank the reviewer for the helpful comment. We have updated the references following the reviewer's suggestion. We have removed the old or not-related references (1, 2, 5, 6, 7, and 8 in the original manuscript) and replaced with new ones (Refs. 1, 2, 4, 5, 7, 8, and 28 in our revised manuscript).

Response to reviewer 2

Comments 1:

There should be a control of the gel with a scrambled shRNA-GFP sequence. In RNA gene silencing work, scrambled sequences are important to verify that there are no off-target effects being observed.

Our response: We thank the reviewer for the helpful comment. We totally agreed to the reviewer's opinion and have verified if there are any off-target effects being observed by monitoring the GFP expression in the presence of 1) scrambled shRNA, 2) plasmid mixtures which can be transcribed into scrambled shRNA, and 3) DNA hydrogel composed of X-DNA and plasmid mixtures which can transcribed into scrambled shRNA. According to the results from the experiments, any significant reduction of GFP expression has not been observed from all three experiments as shown in Figure R1a, b. and the results confirmed that there is no off-target interference effect of scramble shRNA.

In our revised manuscript:

pp5-6

Here, to evaluate the interfering efficiency of the I-gel, four separated sets of expression have been performed by adding different plasmid components to the cell lysate; 1) no additional plasmid as a blank control, 2) GFP plasmid only as a positive reference, 3) GFP plasmid and scrambled shRNA 1 eq (means the copy number of added shRNA is the same as that of I-plasmid), 4) GFP plasmid and scrambled shRNA 100 eq (means the copy number of added shRNA is 100 times of I-plasmid), 5) GFP plasmid and scrambled free plasmid mixture which can produce scrambled shRNA, 6) GFP plasmid and DNA hydrogel which are composed of X-DNA and scrambled shRNA producible plasmid mixture, 7) GFP plasmid and I-plasmid mixture to measure the inhibition of free plasmid, and 8) GFP plasmid and I-gel to see the interfering ability of the gel.

In addition, the results from controls with scrambled shRNA and plasmids mixture expressing scrambled shRNA have proved that there have been no off-target effects of mismatched RNAs on the interfering GFP expression except the perfectly matched shRNA transcribed from free- or gel-phased I-plasmid (Fig. 2a, b).

Comments 2:

The low knockdown with lipofectamine + plasmid is surprising. Did the authors work on optimizing the lipofectamine experimental conditions?

Our response: The reviewer's concern is well taken. In this work, we have not optimized the lipofectamine experimental condition but we have followed the company's instruction manual for the experiment. According to the results of our newly performed experiments (Figure R8a, R13, and R14a), both shRNA and I-plasmid complexed with Lipofectamine have shown higher cellular uptake compared with naked shRNA or free I-plasmid. In case of Lipofectamine-I-plasmid, the cellular uptake measured by fluorescence image analysis has shown

similar efficiency with I-gel. Based on the results, the low knockdown efficiency with lipofectamine + plasmid seems to be attributed to the low transcription efficiency of plasmid in the live cell.

We have added the additional results and revised the manuscript accordingly.

In our revised manuscript:

p11

According to the images taken after incubation of MDCK-GFP cells with Dye-conjugated I-gel (Cy5) in Dulbecco's Modified Eagle Medium (DMEM) at 37 °C under 5% CO₂, high accumulation of I-gel was observed inside cells (Fig. 4a). According to the comparison of cellular uptake efficiency with other controls, free I-plasmid has shown little cellular uptake and I-plasmid which has been complexed with lipofectamine has shown similar cellular uptake efficiency with I-gel. In case of shRNA, the trend of cellular uptake of naked and complexed lipofectamine has been consistent with that of I-plasmid (Supplementary Fig. S8 and Fig.S9a). These results imply that the I-gel can be transported and accumulated in the cell as efficient as a complexed form with Lipofectamine.

Comments 3:

The results shown in figure 3 (qPCR) require a control. Did the authors run a standard for quantification of their samples, or an internal housekeeping gene as a control? The data cannot be normalized in this way (panels c and f) without appropriate controls / quantification.

Our response: We thank the reviewer for the helpful comment. We have performed RT-qPCR (reverse transcription quantitative polymerase chain reaction) to quantify the RNA amount in every cell lysate. Following the reviewer's suggestion, in the quantitation, we have calibrated the amount of RNA using standard samples. The shRNA standard sample has been commercially synthesized and the mRNA standard has been extracted from transcribed solution. For more accurate quantification, a spike-in control RNA (cel-miR-39) has been employed for every shRNA and mRNA RT-qPCR. By adding the spike-in control to the expression lysate before extraction, extraction rates for RNA samples from the lysate could be calculated (Table R1, R2). As shown in Figure R2b, 18 ± 3 and 131 ± 9 copies of shRNA have been produced from a single template free I-plasmid and I-gel, respectively. Under same amount of GFP plasmid (1000 ng), 63 ± 5 and 8 ± 1 copies of GFP mRNA have been produced (Figure R2e) when it is co-expressed with 57 ng of free I-plasmid and I-gel including the same amount of I-plasmid, respectively. These data imply that the transcription efficiency of I-gel is 7.3 folds of free I-plasmid and the interference effect of I-gel is around 7.9 times higher than that of free I-plasmid.

We have added the data and revised the manuscript accordingly.

In our revised manuscript:

pp9-10

In the quantitation, we have calibrated the amount of RNA using standard samples (Supplementary Fig. S7). The

shRNA standard sample has been commercially synthesized and the mRNA standard has been extracted from transcribed solution. For more accurate quantification, a spike-in control RNA (cel-miR-39) has been employed for every RNA RT-qPCR. By adding the spike-in control to the expression lysate before extraction, extraction rates for RNA samples from the lysate could be calculated (Supplementary Table S1 and Table S2). As shown in Figure 3b, when 57 ng of I-plasmid has been used, 18 ± 3 and 131 ± 9 copies of shRNA have been produced from a single template of free I-plasmid and I-plasmid in the I-gel, respectively. In addition to the comparison of siRNA amount from I-gel and plasmid, mRNA amount transcribed from GFP plasmid in the presence of I-gel and free I-plasmid has been evaluated to confirm the reduced GFP fluorescence intensity are mainly due to the interfered mRNA level. The relative amount of GFP mRNA has been also measured by following the same process as shRNA. As seen in the gel electrophoresis analysis, it can be identified that only one gene product was amplified (Fig. 3d) and it is clear that cDNA amount converted from GFP expression lysate in the presence of I-gel is much lower than those from GFP expression lysate with and without I-plasmid. Under same amount of GFP plasmid (1,000 ng), 63 ± 5 and 8 ± 1 copies of GFP mRNA have been produced from a single template GFP plasmid when it is co-expressed with 57 ng of free I-plasmid and I-gel including the same amount of I-plasmid, respectively. (Fig. 3e).

Comments 4:

The entire article needs grammatical corrections.

Our response: We thank the reviewer for the helpful comment. We have re-checked the gramma of the article and tried to correct mistakes. However, if the writing still needs more corrections, we will ask a help of commercial correction service after the article is accepted.

Comments 5:

The authors should cite other works that have looked at siRNA delivery using DNA vehicles. For example, there is a new article from Ren, et al., Nature Communications:

<https://www.nature.com/articles/ncomms13580>. Also, what about other RNA-DNA particles that have been studied? How does your work compare / contrast from other technologies?

Our response: We thank the reviewer for the helpful comment. Following the reviewer's suggestion, we have added a few more recent publications on siRNA delivery using DNA vehicles including the mentioned Nature Communications paper. The originality of our work described in the manuscript is quite unique compared with other previous reports in a view of delivered object. In other works, DNA has been mainly used for delivery of pre-synthesized siRNA. In this case of siRNA delivery, only the amount of delivered siRNA has interference effect. In our concept, however, I-gel behaves as a factory which produces shRNA as well as a vehicle. Also because the transcription carried out on the I-gel produced 131 copies of shRNA per template plasmid (see Figure R2b), there has been an amplification effect of shRNA amounts compared with direct delivery of pre-synthesized shRNA. More thorough studies about the I-gel and comparison with other siRNA delivery vehicles will be

performed and reported in a following paper.

We have cited the literature and added discussion in our revised manuscript.

In our revised manuscript:

p16-17

In this paper, we have, for the first time, demonstrated a hydrogel type of plasmid, I-gel, which is made by simple crosslinking with branched DNAs can make a high performance RNAi reagent. In this newly proposed concept, I-gel has been delivered into a cell and has shown remarkable RNAi effect to silence GFP expression. Comparing with previously reported DNA vehicles for siRNA delivery where DNA has been mainly used for delivery of pre-synthesized siRNA³³, our I-gel is unique in a view of delivered object and takes advantages of transcribing process as an amplification step. In the case of pre-synthesized siRNA delivery, only the amount of delivered siRNA has interference effect. In our concept, however, I-gel behaves as a factory which produces shRNA as well as a vehicle. In addition, because the transcription carried out on the I-gel produced 131 copies of shRNA per template plasmid, there has been an amplification effect of shRNA amounts compared with direct delivery of pre-synthesized shRNA. Still thorough studies about the I-gel and comparison with other siRNA delivery vehicles are needed to improve the applicability of the I-gel. Nonetheless, since the shRNA producing gel has superior stability, cellular uptake efficiency, and silencing effect to the free and lipofectamine complexed plasmid, we expect that I-gel has a huge potential to be used for virus free and also toxicity free RNAi method which is alternative to direct delivery of siRNA.

Comments 6:

Have you measured the gel stiffness, inter-plasmid distance, compactness of dna network (p. 7 line 145) of these gels? If so these general relationships would be important to mention in this paper as well, particularly if it is believed that they impact the siRNA production. If not, obtaining some information with respect to these could be valuable for this work.

Our response: We thank the reviewer for the helpful comment. Some basic physical information about DNA hydrogels including tensile strength, ultimate elongation, DNA incorporation and swelling ratio as well as the transcription efficiency enhancement of crosslinked plasmids have been reported in previous literatures (Nature Materials **5**, 797-801, 2006; Nature Materials **8**, 432-437, 2009). Although we agree that obtaining some physical and conformational information of the I-gel could be valuable, we could not measure the specific properties of I-gel because it has not feasible to measure the properties of nano-sized objects. So we have removed the speculation-based discussion of the effect of compactness of DNA network from our revised manuscript.

Comments 7:

Page 6, line 125: refrain from using the phrase 'so huge'. Replace with more scientific word/ phrase.

Our response: We thank the reviewer for the helpful comment. Following the reviewer's suggestion, we have

replaced the sentence “And the difference of fluorescence intensity was so huge that the difference” with more quantitative one “The four times of fluorescence intensity difference”.

Comments 8:

Page 7, line 147: what do you mean by 57 ng of “genes”? Is that the weight of the linearized i-plasmid estimated to be in there? Should clarify.

Our response: We thank the reviewer for the helpful comment. Following the reviewer’s suggestion, we have replaced “genes” with “linearized I-plasmids”.

Comments 9:

It is not apparent how the cytotoxicity studies were run, specifically how the I-gel and cells were co-incubated. More details are needed here for repeatability.

Our response: We thank the reviewer for the helpful comment. Per the request, we altered the text in the revised manuscript for detail information to clarify experimental condition.

In our revised manuscript:

p25

A MDCK-GFP cell suspension (5,000 cells/well) was dispensed in a 96-well plate (Corning) and incubated for 1 day at 37 °C under 5% CO₂. With the cells attached to the plate, they were co-incubated in DMEM (HyClone) growth media (v/v 10% FBS and 1% penicillin) and dark condition with different concentration of I-gel which is corresponding of X-DNA concentration. I-gel was composed of molar ratio of 1500 X-DNA and one I-plasmid. These samples were incubated for 6, 24, 48 h at 37 °C under 5% CO₂. At the end of each incubation time, Cell Counting Kit-8 solution (CCK-8, Dojindo Laboratories) was added to the samples according to the manufacturer’s instructions.

Comments 10:

The media that the cells were cultured in is not clear. Was it always serum-free DMEM or was there ever serum, were any antibiotics ever used, etc.?

Our response: We thank the reviewer for the helpful comment. In describing the use of serum free media, only v/v 1% of penicillin were used in basic media. When the cells needed to grow and preserve, the growth media was used with v/v 10% FBS (HyClone) and 1% penicillin (HyClone). Per request, we altered the text in the revised manuscript for detail information to clarify experimental condition.

In our revised manuscript:

p22

In case of cell only (control), the cells were co-incubated with DMEM serum free media with v/v 1% penicillin

(HyClone). All the samples washed out with PBS buffer (0.1 M, pH 7.4) for 2 times after 4 hours co-incubation and the cells were additionally incubated for 6 hours with growth media (v/v 10% FBS (HyClone) and 1% penicillin) to ensure cellular recovery and uptake time. The cultured cells were fixed with 4% formaldehyde for 10 min at room temperature and further washed with PBS buffer (0.1 M, pH 7.4) for 3 times. For microscope imaging, cover slips attached the cells were mounted on glass slides using aqueous mounting medium with an anti-fading agent. The fluorescence images were recorded using a Zeiss Axioplan 2 microscope. The pictures were taken using a Zeiss Axiocam HR camera.

pp23-24

First, I-gel at concentration of 262.5 μ M X-DNA and 175 nM I-plasmid was incubated with 300 U T7 RNA polymerase (Thermo Fisher Scientific) for 15 min at room temperature. After incubation, the I-gel sample was treated and diluted to 100 times with DMEM (HyClone) serum free media with 1% penicillin (antibiotics). MDCK-GFP cells had been plated for 24 h in 24 well plates with cover-slip (20,000 cells per each well). The 1/6 times volume of I-gel/polymerase complex was added to each well in 24-well plate of the MDCK-GFP cells. For 'I-plasmid', 'scrambled plasmid mixture', and 'scrambled plasmid gel' set, MDCK-GFP cells were co-cultured with the same molar amount of each plasmid and T7 RNA polymerase as in the case of I-gel. In case of measuring the RNA interference effect of 'naked shRNA' and 'Lipofectamine-shRNA', 100-fold of each RNA sample was used than the number of I-plasmid template considering the shRNA expression rate of I-gel calculated from the cell lysate experiment. For 'naked shRNA' and 'scrambled shRNA' set, MDCK-GFP cells were co-incubated with 175 nM RNA sample. The Lipofectamine-derivative samples were prepared following the manufacturer's instructions. For 'Lipofectamine-I-plasmid' and 'Lipofectamine-scrambled plasmid mixture' sets, Lipofectamine and each plasmid sample were electrostatically complexed by mixing them together in a molar ratio of 5:1 to be final solution concentration of 8.75 nM Lipofectamine and 1.75 nM plasmid substrate. For 'Lipofectamine-shRNA' sets, Lipofectamine shRNA were electrostatically coupled by mixing Lipofectamine and free shRNA in a molar ratio of 5:1 to be final solution concentration of 875 nM Lipofectamine and 175 nM shRNA. The MDCK-GFP cells were co-incubated in serum free media for 4 hours with as-prepared Lipofectamine derivative samples. For 'cell only' set, only MDCK-GFP cells were incubated in serum free media for 4 hours without any samples. All the samples washed out after 4 hours co-incubation and the cells were additionally incubated for 48 hours with DMEM growth media (v/v 10% serum and 1% antibiotics) to evaluate the RNA silencing effect at 37°C under 5% CO₂. The cells are fixed with 4% formaldehyde for 10 min at room temperature and washed with PBS buffer (0.1 M, pH 7.4). For microscope imaging, cover slips attached the cells were mounted on glass slides using aqueous mounting medium with an anti-fading agent.

p25

A MDCK-GFP cell suspension (5,000 cells/well) was dispensed in a 96-well plate (Corning) and incubated for 1 day at 37°C under 5% CO₂. With the cells attached to the plate, they were co-incubated in DMEM (HyClone)

growth media (v/v 10% FBS and 1% penicillin) and dark condition with different concentration of I-gel which is corresponding of X-DNA concentration. I-gel was composed of molar ratio of 1,500 X-DNA and one I-plasmid.

Comments 11:

There is no mention of statistical analysis in the report.

Our response: We thank the reviewer for the helpful comment. The whole data in this manuscript has been triplicated and the average value has been taken as a result. And error bar has been also indicated in each graph or value. We have added the statement in the text and figure legend of our revised manuscript to clarify the statistical analysis.

Response to reviewer 3

Comments 1:

According to Figure 2, I-gel significantly decreased GFP fluorescence intensity while I-plasmid did not affect at all. The authors should clearly describe why they show a difference in the interfering efficiency. The transcription efficiency of I-plasmid should be similar to that of I-gel (I-plasmid + X-DNA) in the presence of the RNA polymerase in the cell lysate assay because siRNA is produced from I-plasmid in the I-gel. Thus, gene knockdown efficiency would be expected to be similar to each other. Does X-DNA in I-gel play special role in transcription?

Our response: We thank the reviewer for the helpful comment. Although it is true that shRNA is produced from I-plasmid in I-gel (I-plasmid crosslinked with X-DNA), the transcription efficiency of I-plasmid alone is much lower than that of I-gel as shown in Figure R4 (in two hours of transcription which has been used for our cell lysate experiments, the shRNA amount produced from I-gel is around 10 times higher than that from I-plasmid) and accordingly the interference efficiency is also lower. The higher transcription efficiency of DNA hydrogel consistently well matched with the results reported previously (Nature Materials **8**, 432-437, 2009). Although the mechanism is not completely understood yet, stability of I-gel might play most important role in enhanced transcription efficiency (Figure R4). As shown in Figure R15, significant amount of I-gel has still remained after 60 minutes under DNase digestion while I-plasmid degraded almost completely in 10 minutes. These results suggested that crosslinked structures of X-DNAs might have protected I-plasmid from DNase attack.

We have added results of the I-gel stability and higher transcription efficiency on our revised manuscript.

In our revised manuscript:

p14-15

As shown in Supplementary Figure S14, the I-plasmid in the solution has almost completely digested in 10 minute of incubation. In case of I-gel, on the contrast, around 90% of the gel has remained even after 60 minute of incubation with DNase (measured using ImageJ). In addition to the high stability of I-gel, transcription efficiency of I-gel has been also higher than that of free I-plasmid (Supplementary Fig. S15). According to the time series comparison of transcription efficiency of free I-plasmid and I-gel, shRNA production from I-gel has increased until 120 minutes, decreased since then while free I-plasmid produced highest amount of shRNA at the initial period, and decreased gradually as time went on. This transcription profile of free I-plasmid and I-gel has been well correlated with the stability of each format of I-plasmid in a free- and a gel-format.

Comments 2:

How stable the T7 RNA polymerase would be in the I-gel and how long it is active in the I-gel. Because stability and activity of the polymerase would largely influence the transcription efficacy in this system, the

authors should provide supporting results.

Our response: We thank the reviewer for the helpful comment. According to our results of monitoring the transcription profile with time in a cell lysate (see Figure R4), the amount of shRNA produced from I-gel increased until 120 minutes and then gradually decreased. However, the shRNA produced from I-plasmid showed the highest amount at the initial time period and then decreased slowly. Since the T7 RNA polymerases have not been encapsulated in I-gel matrix but they simply co-existing with I-gel as well as I-plasmid in the cell lysate or, for the cell experiments, co-incubated with I-gel and I-plasmid before adding to the cells, there is no reason of higher stability of T7 RNA polymerase in I-gel compared with in I-plasmid. Hence, the difference of transcription profile might be mainly due to the transcription efficiency and stability difference between I-gel and I-plasmid not the stability issue of T7 RNA polymerase.

We have added results of the I-gel stability and higher transcription efficiency on our revised manuscript.

In our revised manuscript:

p14-15

As shown in Supplementary Figure S14, the I-plasmid in the solution has almost completely digested in 10 minute of incubation. In case of I-gel, on the contrast, around 90% of the gel has remained even after 60 minute of incubation with DNase (measured using ImageJ). In addition to the high stability of I-gel, transcription efficiency of I-gel has been also higher than that of free I-plasmid (Supplementary Fig. S15). According to the time series comparison of transcription efficiency of free I-plasmid and I-gel, shRNA production from I-gel has increased until 120 minutes, decreased since then while free I-plasmid produced highest amount of shRNA at the initial period, and decreased gradually as time went on. This transcription profile of free I-plasmid and I-gel has been well correlated with the stability of each format of I-plasmid in a free- and a gel-format.

Comments 3:

The authors need to explain how shRNA can be released from the I-gel and show the release profile. ShRNA can be physically captured in the I-gel. ShRNA release amount would affect subsequent gene silencing efficiency.

Our response: We thank the reviewer for the helpful comment. In this experiment, we have employed I-gel of which size is around 100 nm. As mentioned in earlier response, the T7 RNA polymerases have not been encapsulated inside the I-gel and we expect that the polymerases exist out of the gel and also the polymerization mainly happened on the surface of the gel. So little of the shRNA produced might be captured in the I-gel.

Comments 4:

It would be better if qPCR result of in vitro cell lysate system (Figure 3) is combined with Figure 2. In particular, Ct curves (Figure 3b) can move to Supplementary information.

Our response: We thank the reviewer for the helpful comment. Following the reviewer's suggestion, we have

moved the Ct curves to Supplementary information (Supplementary Fig. S7). In case of other data in Fig.3 in the manuscript, we have added more detailed quantitation data and it still remains as an independent figure in our revised manuscript (Fig. 3).

Comments 5:

On page 9, line 197, if the authors want to discuss the intracellular uptake mechanism of the I-gel, they should show the evidence. In the current manuscript, not only the mechanism, but also stability and RNA production of the I-gel were only based on the authors' speculation.

In Figure 4a, if the authors would like to highlight I-gel as an efficient gene carrier in terms of intracellular uptake, a positive control such as lipofectamine is required for comparison.

Our response: We thank the reviewer for the helpful comment.

For the intracellular uptake of the I-gel, we have removed the sentences regarding the intracellular uptake mechanism of the I-gel. Instead, we have performed additional experiments to show the high intracellular uptake efficiency of the I-gel. According to the results of our newly performed experiments (Figure R8a, R13, and R14a), both shRNA and I-plasmid complexed with Lipofectamine have shown higher cellular uptake compared with naked shRNA or free I-plasmid. In case of Lipofectamine-I-plasmid, the cellular uptake measured by fluorescence image analysis has shown similar efficiency with I-gel. Based on the results, the low knockdown efficiency with lipofectamine + plasmid seems to be attributed to the low transcription efficiency of plasmid in the live cell.

In terms of stability and RNA production issue, as shown in Figure R15, the I-gel is much more resistant to DNase digestion compared with the I-plasmid and accordingly high RNA production efficiency has been shown for I-gel.

We have added the additional results and revised the manuscript accordingly.

In our revised manuscript:

p11

According to the images taken after incubation of MDCK-GFP cells with Dye-conjugated I-gel (Cy5) in Dulbecco's Modified Eagle Medium (DMEM) at 37 °C under 5% CO₂, high accumulation of I-gel was observed inside cells (Fig. 4a). According to the comparison of cellular uptake efficiency with other controls, free I-plasmid has shown little cellular uptake and I-plasmid which has been complexed with lipofectamine has shown similar cellular uptake efficiency with I-gel. In case of shRNA, the trend of cellular uptake of naked and complexed lipofectamine has been consistent with that of I-plasmid (Supplementary Fig. S8 and Fig.S9a). These results imply that the I-gel can be transported and accumulated in the cell as efficient as a complexed form with Lipofectamine.

p14

As shown in Supplementary Figure S14, the I-plasmid in the solution has almost completely digested in 10 minute of incubation. In case of I-gel, on the contrast, around 90% of the gel has remained even after 60 minute of incubation with DNase (measured using ImageJ). In addition to the high stability of I-gel, transcription efficiency of I-gel has been also higher than that of free I-plasmid (Supplementary Fig. S15). According to the time series comparison of transcription efficiency of free I-plasmid and I-gel, shRNA production from I-gel has increased until 120 minutes, decreased since then while free I-plasmid produced highest amount of shRNA at the initial period, and decreased gradually as time went on. This transcription profile of free I-plasmid and I-gel has been well correlated with the stability of each format of I-plasmid in a free- and a gel-format.

Comments 6:

In Figure 4b, GFP-siRNA/lipofectamine is more suitable as a control instead of I-plasmid/lipofectamine to compare the gene silencing effect because I-gel contains RNA polymerase unlike I-plasmid/lipofectamine. That means I-plasmid/lipofectamine can produce siRNA only if it is transported into nucleus while I-gel can generate siRNA in the cytosol. In addition, neither free siRNA or shRNA is employed as a control in the current study.

Our response: We thank the reviewer for the helpful comment. Actually, both cases for the free I-plasmid/lipofectamine as well as I-gel have been co-incubated with T7 RNA polymerase before adding to cells. Therefore, it might not be required for free I-plasmid/lipofectamine to be transported into nucleus to produce shRNA in the cell.

Following the reviewer's suggestion, we have treated the cell with free shRNA as well as shRNA/lipofectamine as controls. According to the cell fluorescence images (Figure R16 and R14b) representing GFP expression efficiency, there has been little interference for free shRNA compared with that of no-treated cells while shRNA/lipofectamine has significantly reduced the GFP expression. The results mean that the cellular uptake is essential for the interference effect of shRNA.

We have added the additional results and revised the manuscript accordingly.

In our revised manuscript:

p11

According to the images taken after incubation of MDCK-GFP cells with Dye-conjugated I-gel (Cy5) in Dulbecco's Modified Eagle Medium (DMEM) at 37 °C under 5% CO₂, high accumulation of I-gel was observed inside cells (Fig. 4a). According to the comparison of cellular uptake efficiency with other controls, free I-plasmid has shown little cellular uptake and I-plasmid which has been complexed with lipofectamine has shown similar cellular uptake efficiency with I-gel. In case of shRNA, the trend of cellular uptake of naked and complexed lipofectamine has been consistent with that of I-plasmid (Supplementary Fig. S8 and Fig.S9a). These results imply that the I-gel can be transported and accumulated in the cell as efficient as a complexed form with Lipofectamine.

p14

As shown in Supplementary Figure S14, the I-plasmid in the solution has almost completely digested in 10 minute of incubation. In case of I-gel, on the contrast, around 90% of the gel has remained even after 60 minute of incubation with DNase (measured using ImageJ). In addition to the high stability of I-gel, transcription efficiency of I-gel has been also higher than that of free I-plasmid (Supplementary Fig. S15). According to the time series comparison of transcription efficiency of free I-plasmid and I-gel, shRNA production from I-gel has increased until 120 minutes, decreased since then while free I-plasmid produced highest amount of shRNA at the initial period, and decreased gradually as time went on. This transcription profile of free I-plasmid and I-gel has been well correlated with the stability of each format of I-plasmid in a free- and a gel-format.

Comments 7:

qPCR should be performed to quantify the actual gene knockdown efficiency in living cells. Additional experiments are required.

Our response: We thank the reviewer for the helpful comments. Following the reviewer's suggestion, we have thoroughly studied the shRNA and mRNA amount level in live cells using RT-qPCR. According to the results, GFP mRNA level in I-gel treated cell has shown about 25% of that transcribed in non-treated cell (cell only) and only shRNA complexed with Lipofectamine treated cell has shown similar interference effect with I-gel treated one (Figure R8d). For the case of shRNA, the relative amounts of shRNA in the cell have been measured RT-qPCR and the result have been summarized in Figure R8e)

We have added the additional results and revised the manuscript accordingly.

In our revised manuscript:

P13

For a more strict and quantitative comparison of the interference effect of I-gel against the GFP expression, the GFP mRNA and shRNA in the cells have been quantified using RT-qPCR (Fig. 4d and 4e). As it has been done in lysate experiment, a spike-in control RNA (cel-miR-39) has been employed for every RNA RT-qPCR. As shown in the results, I-gel has interfered the GFP mRNA transcription most effectively in the cells (only one fourth of the GFP mRNA amount was measured compared with non-treated cells). Only shRNA complexed with lipofectamine has shown similar interference effect with the I-gel. In case of shRNA, which has existed in the cells, the shRNA amounts transcribed from the I-gel was 8 times and 2 times higher than that produced from lipofectamine-I-plasmid and remained shRNA that had been added as a complexed form with lipofectamine, respectively.

Figures and Tables:

Figure R1. Optimization and evaluation of I-gel. (a) The FL spectra of GFP in each condition (no plasmid, only GFP expression, addition of scrambled shRNA 1eq, scrambled shRNA 100eq, scrambled plasmid mixture, scrambled plasmid gel, I-plasmid, and I-gel). (b) A comparison of the fluorescence intensities at the peak points of the graph (a) and the inset is fluorescence images of each sample. The relative FL intensity (c) with the addition of increasing amount of I-plasmid and I-gel, (d) with the variation of X-DNA and I-plasmid ratio in I-gel. For all experiments, 57 ng of I-plasmid has been used except (c). Error bars represent standard deviations from three replicates.

Table R1. RNA recovery rates and GFP mRNA concentration in each sample

Sample	CT value (spike-in control)	Recovery Rate(%)	Conc. of GFP mRNA (nM)
ref miR-39	19.37 ± 0.27	100	NA
Only GFP plasmid	19.58 ± 0.05	86.75 ± 2.56	622.0 ± 25.8
GFP plasmid + Scrambled shRNA	19.87 ± 0.04	71.73 ± 1.94	647.5 ± 43.0
GFP plasmid + Scrambled plasmid mix	19.57 ± 0.06	87.32 ± 3.50	593.6 ± 61.6
GFP plasmid + Scrambled I-gel	19.85 ± 0.05	72.99 ± 2.21	625.9 ± 25.0
GFP plasmid + I-plasmid	19.54 ± 0.06	89.25 ± 3.39	588.7 ± 47.6
GFP plasmid + I-gel	20.17 ± 0.04	59.04 ± 1.69	73.4 ± 5.2

Table R2. RNA recovery rates and shRNA concentration in each sample

Sample	CT value (spike-in control)	Recovery Rate(%)	Conc. of GFP mRNA (nM)
ref miR-39	19.37 ± 0.27	100	NA
I-plasmid	20.92 ± 0.08	36.18 ± 1.97	12.7 ± 1.8
I-gel	21.09 ± 0.02	32.22 ± 0.47	91.9 ± 6.5

Figure R2. Comparison of RNA expressions. (a, d) The RT-PCR products in agarose gel electrophoresis for (a) shRNA and (d) GFP mRNA detection in extracted RNA samples from the cell lysates. (a) Samples on gel are lane M: Molecular weight standard marker; Lane 1: standard shRNA, Lane 2: shRNA expressed from I-plasmid, Lane 3: shRNA expressed from I-gel; (d) Lane 1: only GFP plasmid, Lane 2, 3, 4, 5, 6: the addition of scrambled shRNA, scrambled plasmid mixture, scrambled plasmid gel, I-plasmid, and I-gel. (b, e) Copy number of RNA expressed from a template plasmid for (b) shRNA and (e) GFP mRNA expression in each condition. (c, f) The relative RNA expression value for (c) shRNA and (f) GFP mRNA in 5.7 ng, 57 ng and 114 ng of I-plasmid amount condition. The insets of (c) and (f) represent zoomed up images of dashed box. Error bars represent standard deviations from three replicates.

Figure R3. GFP expression with increased I-plasmid amount. (a) The normalized FL spectra, (b) the ratio of peak intensity of expressed GFP with increased amount of I-plasmid (0, 150, 300, 500, 1000, and 2000 ng). Error bars represent standard deviations from three replicates.

Figure R4. Comparison of transcription efficiency of I-gel and I-plasmid. Time series monitoring of shRNA production. Error bars represent standard deviations from three replicates.

Figure R5. The siRNA expression plasmid (I-plasmid) map and expected shRNA transcription. The designed sequence was inserted the site between BbsI and BamHI. This plasmid maxi-prepped for amplification and linearized by BamHI restriction enzyme. The restriction site is complementary and palindromic sequence with each X-DNA and plasmid itself. The X-DNAs and linearized plasmids can be linked by their sticky-ends. The I-gel can be synthesized with covalently conjugation of sticky-end through ligase enzyme reaction.

Figure R6. The measured melting points of RT-PCR products. Blue: from standard shRNA, yellow: from free I-plasmid, and red: from I-gel. Error bars represent standard deviations from three replicates.

Figure R7. (a) The relative GFP FL intensity with the addition of increasing amount of I-plasmid and I-gel. (b, c) The relative RNA expression value for (b) shRNA and (c) GFP mRNA in 5.7 ng, 57 ng and 114 ng of I-plasmid amount condition. The insets of (b) and (c) represent zoomed up images of dashed box. Error bars represent standard deviations from three replicates.

Figure R8. Gene-silencing effect by I-gel in a cellular level. (a) Comparison of intracellular uptake of plasmids in different formats. Second from left: I-plasmid labeled with Cy5, third: I-plasmid labeled with Cy5 complexed with lipofectamine, and the last: I-gel labeled with Cy5. Scale bar: 20 μ m. (b) Fluorescence images of MDCK GFP-expressing (MDCK-GFP) cells co-incubated with I-plasmid, I-plasmid complexed with lipofectamine, and I-gel. Scale bar: 20 μ m. (c) Fluorescence-activated cell sorter analysis of GFP-expressing cell line, MDCK-GFP cells after treatment of sample/polymerase complexes in serum-deficient medium. The number means the percentage of GFP-overexpressing cells sorted within a pre-fixed gate region as indicated by a bar. (d, e)

The relative amounts of (d) GFP mRNA and (e) shRNA from GFP expressing MDCK cells after incubation in various conditions. (*used RNA amounts were 100-fold to the number of plasmid considering the RNA transcription rate of the I-gel). Error bars represent standard deviations from three replicates.

Figure R9. Agarose gel electrophoresis image of the I-gel in various conditions. Note: DNA was resolved on 2% agarose gel stained with 1x Gel-red. Samples on gel are lane M: DNA marker (25~2,000 bps); lane 1: 36 base single strand DNA (one element composing of X-DNA); lane 2: 3 sticky-ends X-DNA; lane 3~9: 1000, 1500, 2000, 3000, 4000, 5000, 6000 X-DNA and I-plasmid ratio of nano-scale I-gel; lane 10: I-plasmid.

Figure R10. Agarose gel electrophoresis profile of RT-PCR control about GFP mRNA, shRNA experiments. Each sample is amplified using specific primers for (a) siRNA or (b) mRNA of GFP detection in extracted RNA samples from the cell lysate system. Note: DNA was resolved on 2% agarose gel for 1h (a) lane M: Molecular weight standard marker; lane P: I-gel positive control; lane 1: no DNA; lane 2: only GFP plasmid; lane 3: only scrambled siRNA; lane 4: GFP plasmid and scrambled siRNA; lane 5: PCR negative; lane 6: no RT only GFP plasmid; lane 7: no RT only I-gel; lane 8: no RT only I-plasmid; lane 9: no RT only scrambled siRNA; lane 10: no RT GFP plasmid and I-gel; lane 11: no RT GFP plasmid and I-plasmid; lane 12: no RT GFP plasmid and scrambled siRNA; lane 13: scrambled I plasmid; lane 14: scrambled I gel; lane 15: GFP plasmid and scrambled I plasmid; lane 16: GFP plasmid and scrambled I gel; lane 17 no RT scrambled I plasmid; lane 18 no RT scrambled I gel; lane 19: no RT GFP plasmid and scrambled I plasmid; lane 20 no RT GFP plasmid and scrambled I gel. (b) lane M: Molecular weight standard marker; lane P: GFP positive control; lane 1: no DNA; lane 2: only I-gel; lane 3: only I-plasmid; lane 4: only scrambled siRNA; lane 5: PCR negative; lane 6: no RT only GFP plasmid; lane 7: no RT only I-gel; lane 8: no RT only I-plasmid; lane 9: no RT only scrambled siRNA; lane 10: no RT GFP plasmid and I-gel; lane 11: no RT GFP plasmid and I-plasmid; lane 12: no RT GFP plasmid and scrambled siRNA; lane 13: scrambled I plasmid; lane 14: scrambled I gel; lane 15: no RT scrambled I plasmid; lane 16: no RT scrambled I gel.

siRNA; lane 4: GFP plasmid and scrambled siRNA; lane 5: PCR negative; lane 6: no RT only GFP plasmid; lane 7: no RT only I-gel; lane 8: no RT only I-plasmid; lane 9: no RT only scrambled siRNA; lane 10: no RT GFP plasmid and I-gel; lane 11: no RT GFP plasmid and I-plasmid; lane 12: no RT GFP plasmid and scrambled siRNA; lane 13: scrambled I plasmid; lane 14: scrambled I gel; lane 15: GFP plasmid and scrambled I plasmid; lane 16: GFP plasmid and scrambled I gel; lane 17 no RT scrambled I plasmid; lane 18 no RT scrambled I gel; lane 19: no RT GFP plasmid and scrambled I plasmid; lane 20 no RT GFP plasmid and scrambled I gel. (b) lane M: Molecular weight standard marker; lane P: GFP positive control; lane 1: no DNA; lane 2: only I-gel; lane 3: only I-plasmid; lane 4: only scrambled siRNA; lane 5: PCR negative; lane 6: no RT only GFP plasmid; lane 7: no RT only I-gel; lane 8: no RT only I-plasmid; lane 9: no RT only scrambled siRNA; lane 10: no RT GFP plasmid and I-gel; lane 11: no RT GFP plasmid and I-plasmid; lane 12: no RT GFP plasmid and scrambled siRNA; lane 13: scrambled I plasmid; lane 14: scrambled I gel; lane 15: no RT scrambled I plasmid; lane 16: no RT scrambled I gel.

Figure R11. Agarose gel electrophoresis image of the I-gel and its ingredients.

Note: DNA was resolved on 2% agarose gel stained with 1x Gel-red. Samples on gel are lane M: DNA marker (25~2,000 bps); lane 1: 36 base single strand DNA (one element composing of X-DNA); lane 2: 3 sticky-ends X-DNA; lane 3: I-gel; lane 4: Blank-gel; lane 5: shRNA expressing plasmid (I-plasmid).

Figure R12. SEM images of I-gel. SEM images of large scale I-gel and nano-scale I-gel at the ratio of 1,500:1 of X-DNA and plasmid.

Figure R13. Comparison of intracellular uptake of shRNA in different formats. First and second from left: negative controls, cell only and non-labeled I-gel, third: shRNA labeled with Cy5, and the last: lipofectamine complexed shRNA labeled with Cy5. Scale bar: 20 μ m..

Figure R14. The relative pixel intensities from (a) Cy5 and (b) GFP fluorescence images of non-treated MDCK-GFP cell (cell only), and treated cells with I-plasmid, I-plasmid complexed with Lipofectamine, and I-gel. Error bars represent standard deviations from three replicates.

Figure R15. Stability of I-plasmid and I-gel. Gel electrophoresis of I-plasmid (a) and I-gel (b) treated with DNase 1 at 37°C, lane 1: 0min, lane 2: 1min, lane 3: 3min, lane 4: 5min, lane 5: 10min, lane 6: 30min, lane 7: 60min.

Figure R16. Fluorescence images of MDCK GFP-expressing (MDCK-GFP) cells co-incubated with only shRNA and Lipofectamine-shRNA complex. (a) FL Images in condition of only shRNA and Lipofectamine-shRNA complex for positive control of RNA interference effect. (b) FL Images in condition of only scrambled siRNA, only scrambled plasmid mixture, Lipofectamine-plasmid mixture complex and scrambled plasmid gel for negative control of RNA interference effect. Scale bar: 50 μ m.

Please address all correspondence to:

Nokyoung Park

Department of Chemistry

Myongji University

116 Myongji-ro, Cheoin-gu, Yongin, South Korea

Phone: +82-31-330-6188

Email: pospnk@mju.ac.kr

Thank you again for considering our work for publication, and please do not hesitate to contact us if you have any questions or require additional information.

Sincerely,

Authors

Reviewers' comments:

Reviewer #1 (Remarks to the Author):

In the paper was largely revised. It is the opinion of this reviewer that the data is still not convincing and the work is too preliminary to be accepted for publication.

Detailed comments:

Wrong reference: P3, l38 „nanorobots and used as templates for precise assemblies of inorganic materials6“. – Ref 6 does not deal with inorganic materials

Fig.2: Caption should indicate that this data was obtained by in vitro transcription/translation with HeLa cell extract. It is still very surprising that siRNA or siPlasmid do not show any reduction in protein expression.

Fig.3a: How were the samples for the gel prepared? Lanes 2 and 3 show bands of (at most) equal intensity – how does this correlate with the proposed substantial higher shRNA production from the I-gel?

Fig. 3d: what exactly is shown here – cDNA? How exactly were the samples prepared? Here we (presumably) see an effect of the I-gel on the band intensity, while this is lacking in Fig3a – explanation?

Fig. S4: It is unclear what this figure should indicate. No bands are visible in the gel except very weak ones for the positive control

The quantitative analysis leads to confusing results:

Fig. R3b indicates that 2000 ng of I-plasmid are necessary to produce a similar inhibitory effect than 53 ng of I-gel

⇒ approx. 40 fold higher transcription rate for I-gel

P8, line 161: „RNAi efficiency of the gel was 57.6 times higher than free I-plasmid at this condition (Fig. 2c and 2d)“

⇒ approx. 55 fold higher transcription rate for I-gel

P9, line 202: „ 8 ± 3 and 131 ± 9 copies of shRNA have been produced from a single template of free I-plasmid and I-plasmid in the I-gel, respectively“.

⇒ approx. 15 fold higher transcription rate for I-gel

P10, line 211: „ 63 ± 5 and 8 ± 1 copies of GFP mRNA have been produced from a single template GFP plasmid when it is co-expressed with 57 ng of free I-plasmid and I-gel including the same amount of I-plasmid, respectively“

⇒ approx. 8 fold higher inhibition rate for I-gel

P10, line 223: It is stated that „When 114 ng of I-plasmid 224 has been used, however, the reduction of fluorescence intensity (interference effect) of free I-225 plasmid is 22.5% which is around 3 times lower than that of I-gel, 70.1%. The three times 226 lower reduction rate of free I-plasmid has been attributed to the 4.5 times lower transcription 227 rate of free I-plasmid compared with I-gel“ – Which data is this statement referring to? This reviewer is unable to follow the calculations.

Fig. 5: The relative amounts of (d) GFP mRNA and (e) shRNA from GFP-expressing MDCK cells differ substantially when one compares lipo-shRNA and I-gel. Explanation? In the best case (Fig. 5e) one could extract a two-fold improvement for the I-gel.

Altogether, the quantitative analysis leads to confusing results which do not give a consistent picture of what might actually be happening in the experiments. Furthermore, the authors stress that the I-gel resembles a „synthetic nucleus“ that produces the siRNA. To strengthen this hypothesis, it would be necessary to have more information about the gel uptake and its fate inside the cell. This could be easily achieved with the fluorescently labeled X-DNA. An important question to answer is how stable the gels are inside the cell.

Reviewer #2 (Remarks to the Author):

The authors have substantially revised the manuscript and satisfactorily responded to the comments. I have no further comments, except that the grammar still needs correcting throughout the manuscript.

Editorial note: Reviewer #3 was unable to provide feedback on the revision. Another reviewer with similar expertise was secured and asked to comment on the authors response to reviewer #3, this reviewer is numbered as Reviewer #4.

Reviewer #4 (Remarks to the Author):

The authors have responded to Reviewer 3's comments in detail - however, several clarifying questions still remain and should be addressed.

1. A few things are still speculative - e.g. the authors indicate in their response that they expect the T7 polymerase to remain outside the I-gels when co-incubated and that transcription takes place on the surface of the I-gel only. However the schematic in Figure 1 could be interpreted by readers as T7 polymerase being inside the I-gel (As Reviewer 3 did as well) - this needs to be clearly corrected and clarified. However, what is the evidence that the T7 polymerase (which is 99kd) is not entering the I-gel when co-incubated? Most hydrogels are extremely porous and allows fairly free movement of proteins. If they do go in, it could be an important distinction with Lipofectamine. Some experiments to address this is important - e.g. a simple nanogold labeling of the T7 followed by TEM of the I-gel could show its location. Or super-resolution microscopy could be employed. Regardless - the speculation that they are on the outside only is not supported by any data.

2. In response to reviewer comments the authors performed simple DNase digestion experiments. Although this had been classically employed by the gene delivery community, it's relevance is questionable. Where inside the cell are DNases and at this concentration? If the authors want to demonstrate stability, is there a way to extract the plasmid from the I-gel and run it in the gel and compare with the free plasmid. Most delivery formulations show stability in this manner. In fact a functional assay (transfection) with the extracted plasmid would be better. This hypothesis that the I-gel protects the plasmid better than naked I-plasmid inside the cell remains speculative.

3. A key aspect of the authors response is the comparison with Lipofectamine. Supplementary Fig S9 showed that although similar amount of I-plasmid went into the cells whether it was delivered by I-gel or Lipofectamine. Yet - the GFP expression was much better silenced by I-gel than Lipofectamine. The authors speculated that this because I-gel allowed better transcription than lipofectamine. However, the variance in figure 4e makes it difficult to support this conclusion. At the minimum Fig 4 should show statistical significance in all groups through appropriate analyses. Also a larger number of "n" may mitigate the variance problem seen in Lipofectamine+I-plasmid

mediated shRNA production and allow for a cleaner conclusion.

Response to Reviewers:

Manuscript ID: NCOMMS-17-24306B

Title: A RNA producing DNA hydrogel as a platform for a high performance RNA interference system

Dear Editor,

Thank you so much for the correspondence with the referees. We appreciate all the sincere comments by reviewers on our manuscript. The comments were carefully considered during the revision of the manuscript. In the upcoming pages, we tried to respond to these comments one by one. Our detailed list of changes and responses to reviewers' concerns are given below.

Response to reviewer 1

Comments 1:

Wrong reference: P3, 138 „nanorobots and used as templates for precise assemblies of inorganic materials⁶“. – Ref 6 does not deal with inorganic materials

Our response: We thank the reviewer for the helpful comment. As the reviewer point out, the Ref 6 does not deal with inorganic materials but it does with DNA nanorobot. Therefore, we have updated the references as follows.

In our original manuscript: p3

Moreover, these DNA nanostructures have shown their own structural functions such as nanorobots and used as templates for precise assemblies of inorganic materials⁶.

In our revised manuscript: p3

Moreover, these DNA nanostructures (e.g. nanorobots⁶) have revealed their own structural functions and have been used as templates for the precise assembly of inorganic materials⁷.

Ref 6: A logic-gated nanorobot for targeted transport of molecular payloads. Science 335, 831-834 (2012).

Ref 7: Sol–Gel Reaction Using DNA as a Template: An Attempt Toward Transcription of DNA into Inorganic Materials. Angew. Chem. Int. Ed. 3279-3283 (2004).

Comments 2:

Fig.2: Caption should indicate that this data was obtained by in vitro transcription/translation with HeLa cell extract. It is still very surprising that siRNA or siPlasmid do not show any reduction in protein expression.

Our response: We thank the reviewer for the helpful comment. Following the reviewer's suggestion, we have added a sentence in the caption indicating that the data was obtained by in vitro transcription/translation with HeLa cell extract.

In our revised manuscript:

Figure 2. Optimization and evaluation of I-gel. (a) The FL spectra of GFP in each condition (no plasmid, only GFP expression, addition of scrambled shRNA 1eq, scrambled shRNA 100eq, scrambled plasmid mixture, scrambled plasmid gel, I-plasmid, and I-gel). (b) A comparison of the fluorescence intensities at the peak points of the graph (a) and the inset is fluorescence images of each sample. The relative FL intensity (c) with the addition of increasing amount of I-plasmid and I-gel (*P < 0.01, **P < 0.001 and ***P < 0.0001 are noted between I-plasmid and I-gel), (d) with the variation of X-DNA and I-plasmid ratio in I-gel. For all experiments, 57 ng of I-plasmid has been used except (c). This data was obtained by in vitro transcription/translation with HeLa cell lysate system. Error bars represent standard deviations from three replicates.

For the low interference effect of I-plasmid shown in Figure 2 (the applied amount of free I-plasmid and the I-plasmid in I-gel was same to be 57 ng each), it is thought to be mainly due to the lower transcription rate of I-plasmid compared with that of I-gel as we have shown in Figure 3 and Supplementary Table S2. As shown in Supplementary Figure S2, 2,000 ng of I-plasmid has shown similar interfering effect with that of I-gel with 57 ng of I-plasmid.

Actually, Figure 2 is dealing with indirect effect of shRNA expressed from I-plasmid and I-gel. We have examined the direct interference effect of shRNA only with cells not in the lysate. According to the results (shown in Figure 4d and Supplementary Figure S10 in our manuscript), although free shRNA has shown low GFP mRNA reduction, it has shown similar interference effect with that of I-gel when the shRNA has been delivered in a complexed form with Lipofectamine.

Comments 3:

Fig.3a: How were the samples for the gel prepared? Lanes 2 and 3 show bands of (at most) equal intensity – how does this correlate with the proposed substantial higher shRNA production from the I-gel?

Our response: Point is well taken. Fig. 3a shows DNA bands which have been obtained through 40 cycles of PCR after reverse transcription of shRNAs transcribed from standard shRNA (lane 1), I-plasmid (lane 2), and I-gel (lane 3), respectively. The gel electrophoresis has been run to confirm if the amplified product of DNA is a correct one which was reverse transcribed from each RNA and each band shows similar intensities because the amounts of DNA have been saturated after 40 cycles of PCR. This response on the similar bands intensities of gel images can be also applied to the bands in Fig. 3d in a similar way.

The substantial shRNA production from the I-gel has been demonstrated in Fig. 3b, 3c, 3e, and 3f which have been obtained from RT-qPCR.

We sorry for any inconvenience caused by the insufficient description in the caption and we have revised our manuscript accordingly.

In our revised manuscript:

Figure 3. Comparison of RNA expressions. (a, d) The RT-PCR products in agarose gel electrophoresis for (a) shRNA and (d) GFP mRNA detection in extracted RNA samples from the cell lysates. (a) Samples on gel are lane M: Molecular weight standard marker; **cDNAs reverse transcribed respectively from** Lane 1: standard shRNA, Lane 2: shRNA expressed from I-plasmid, Lane 3: shRNA expressed from I-gel; (d) **cDNAs reverse transcribed respectively from the GFP mRNAs produced in the cell lysates transcribed/translated in the presence of** Lane 1: only GFP plasmid, Lane 2, 3, 4, 5, 6: the addition of scrambled shRNA, scrambled plasmid mixture, scrambled plasmid gel, I-plasmid, and I-gel. (b, e) Copy number of RNA expressed from a template plasmid for (b) shRNA and (e) GFP mRNA expression in each condition (***P < 0.01, **P < 0.001 and ***P < 0.0001 are noted between condition marked by line and I-gel**). (c, f) The relative RNA expression value for (c) shRNA and (f) GFP mRNA in 5.7 ng, 57 ng and 114 ng of I-plasmid amount condition. The insets of (c) and (f) represent zoomed up images of dashed box (***P < 0.01, **P < 0.001 and ***P < 0.0001 are noted between I-plasmid and I-gel**). Error bars represent standard deviations from three replicates.

Comments 4:

Fig. 3d: what exactly is shown here – cDNA? How exactly were the samples prepared? Here we (presumably) see an effect of the I-gel on the band intensity, while this is lacking in Fig3a – explanation?

Our response: Point is well taken and please refer the above response for the explanation.

Comments 5:

Fig. S4: It is unclear what this figure should indicate. No bands are visible in the gel except very weak ones for the positive control

Our response: Point is well taken. The Fig. S4 shows that the designed primers for RT-PCR of shRNA and GFP mRNA do work on each cDNA specifically. Each lane in the image represents the PCR amplified cDNA which has been reverse transcribed from the RNA extracted from cell lysates. Therefore it has been expected that bands are visible only in positive control lanes including the cDNAs amplified with correctly designed primers. Actually, this data has been modified to reflect the reviewer's previous comment 'Each condition requires at least one positive control also to warrant that the RT-PCR worked correctly'.

Since we think Fig. S4 may induce potential inconvenience by the confusing and inappropriate data and the data seems not necessary to support our I-gel concept, we have removed Fig. S4 from Supplementary Information and revised the manuscript accordingly.

In our original manuscript: p8

As can be seen in the image, only one gene product was amplified in each sample, verifying that the RT-PCR had produced only the intended cDNA, as no band was observed from the RT-PCR negative control without template

In our revised manuscript: p8

As can be seen in the image, only one gene product was amplified in each sample, verifying that the RT-PCR had produced only the intended cDNA.

Comments 6:

The quantitative analysis leads to confusing results:

Fig. R3b indicates that 2000 ng of I-plasmid are necessary to produce a similar inhibitory effect than 53 ng of I-gel \Rightarrow approx. 40 fold higher transcription rate for I-gel

P8, line 161: „RNAi efficiency of the gel was 57.6 times higher than free I-plasmid at this condition (Fig. 2c and 2d)“ \Rightarrow approx. 55 fold higher transcription rate for I-gel

P9, line 202: „ 8 ± 3 and 131 ± 9 copies of shRNA have been produced from a single template of free I-plasmid and I-plasmid in the I-gel, respectively“ \Rightarrow approx. 15 fold higher transcription rate for I-gel

P10, line 211: „ 63 ± 5 and 8 ± 1 copies of GFP mRNA have been produced from a single template GFP plasmid when it is co-expressed with 57 ng of free I-plasmid and I-gel including the same amount of I-plasmid, respectively“ \Rightarrow approx. 8 fold higher inhibition rate for I-gel

Our response: Points are well taken. Each result pointed out by the reviewer have represented different conditions of transcription or interference.

1. Fig. R3b indicates that 2,000 ng of I-plasmid are necessary to produce a similar inhibitory effect than 53 ng of I-gel \Rightarrow approx. 40 fold higher transcription rate for I-gel
2. P8, line 161: „RNAi efficiency of the gel was 57.6 times higher than free I-plasmid at this condition (Fig. 2c and 2d)“ \Rightarrow approx. 55 fold higher transcription rate for I-gel
3. P9, line 202: „ 8 ± 3 and 131 ± 9 copies of shRNA have been produced from a single template of free I-plasmid and I-plasmid in the I-gel, respectively“ \Rightarrow approx. 15 fold higher transcription rate for I-gel
4. P10, line 211: „ 63 ± 5 and 8 ± 1 copies of GFP mRNA have been produced from a single template GFP plasmid when it is co-expressed with 57 ng of free I-plasmid and I-gel including the same amount of I-plasmid, respectively“ \Rightarrow approx. 8 fold higher inhibition rate for I-gel

--- In case of 1 and 2, the interference effect had been measured by quantifying the fluorescence of GFP in the cell

lysate.

The 57.6 times difference of case 2 had been obtained from cell lysate samples transcribed/translated in the presence of same amount (57 ng) of I-plasmid in a free or gel format. This condition was optimized for I-gel resulting the maximized interference fold difference.

The experiment of case 1 has been performed to demonstrate that the I-plasmid also able to show significant interference effect if the amount is enough. The results have been obtained in several different amounts of I-plasmid resulting that the interference effect of 2,000 ng I-plasmid showed similar to that of I-gel containing 57 ng of I-plasmid. As the reviewer calculated, the transcription efficiency could be roughly 40 times different. However, we cannot directly calculate the fold increase by dividing the plasmid amounts which are showing similar interference effect because the interference effect was not linearly correlated with the used amount of I-plasmid as shown in Supplementary Figure S2b.

--- In case of 3 and 4, the number of shRNA and mRNA which transcribed from each plasmid has been quantified by employing RT-qPCR. In case of shRNA, 18 ± 3 and 131 ± 9 copies (7.3 fold) of shRNA have been produced from a single template of free I-plasmid and I-plasmid in the I-gel, respectively. In case of mRNA, 63 ± 5 and 8 ± 1 copies (7.9 fold) of GFP mRNA have been produced from a single template GFP plasmid when it is co-expressed with 57 ng of free I-plasmid and I-gel including the same amount of I-plasmid, respectively. According to the new calculation based on the result which had been described in our previously submitted manuscript, the fold difference in case of 3 (7.3 fold) and 4 (7.9 fold) seemed not significant.

Comments 7:

P10, line 223: It is stated that „When 114 ng of I-plasmid has been used, however, the reduction of fluorescence intensity (interference effect) of free I-plasmid is 22.5% which is around 3 times lower than that of I-gel, 70.1%. The three times lower reduction rate of free I-plasmid has been attributed to the 4.5 times lower transcription rate of free I-plasmid compared with I-gel–Which data is this statement referring to? This reviewer is unable to follow the calculations.

Our response: We thank the reviewer for the helpful comment. The statement about the reduction of GFP fluorescence by 22.5% and 70.5% in conditions of 114 ng of I-plasmid and 114 ng of I-gel, respectively refers to the data represented in Fig. 2c. The statement that 114 ng of I-gel have the 4.5-fold higher transcription rate than 114 ng of I-plasmid refers the data in Fig. 3c.

We sorry for any inconvenience and we have revised our manuscript to clearly indicate the referring data.

In our original manuscript: p10

When 114 ng of I-plasmid has been used, however, the reduction of fluorescence intensity (interference effect) of free I-plasmid is 22.5% which is around 3 times lower than that of I-gel, 70.1%. The three times lower reduction rate of free I-plasmid has been attributed to the 4.5 times lower transcription rate of free I-plasmid compared with I-gel.

In our revised manuscript: p10

When 114 ng of I-plasmid was used, however, the reduction of fluorescence intensity (interference effect) by the free I-plasmid was 22.5%, which was around 3-times lower than that of the I-gel (70.1%) (Fig. 2c). The 3-times lower reduction rate of the free I-plasmid was attributed to its 4.5-times lower transcription rate compared with that of the I-gel (Fig. 3c).

Comments 8:

Fig. 5: The relative amounts of (d) GFP mRNA and (e) shRNA from GFP-expressing MDCK cells differ substantially when one compares lipo-shRNA and I-gel. Explanation? In the best case (Fig. 5e) one could extract a two-fold improvement for the I-gel.

Our response: Point is well taken. The reviewer might be mentioning on the Fig. 4d and 4e. We have compared the interference effect of Lipo-shRNA with I-gel by measuring the GFP mRNA and have examined the remained amount of shRNA in both cases after completing the cellular experiment. Therefore, the two fold of shRNA in I-gel case than Lipo-shRNA (Fig. 4e) doesn't mean that the I-gel should have better interference effect through the whole experimental process and the result has showed they have had similar interference effect in our experimental condition (Fig. 4d). Although we don't know the shRNA amount at a specific time point in each case since we have not studied the time profile of shRNA in the cell, we can reasonably expect that the initial amount of shRNA of Lipo-shRNA (87.5 pmol) is extremely higher than that expressed from I-gel (theoretically 0M at time zero) and then shRNA amount should be changed for both cases; only degradation for Lipo-shRNA and degradation/production for I-gel.

To minimize any confusion, we have revised our manuscript accordingly.

In our original manuscript: p13

In case of shRNA, which has existed in the cells, the shRNA amounts transcribed from the I-gel was 8 times and 2 times higher than that produced from lipofectamine-I-plasmid and remained shRNA that had been added as a complexed form with lipofectamine, respectively.

In our revised manuscript: p13

In the case of shRNA, which had existed in the cells, the amounts transcribed from the I-gel was 9- and 2.5-times higher, respectively, than that produced from Lipofectamine-complexed I-plasmid and the amount of remaining shRNA that had been added one time initially as a form complexed with Lipofectamine.

Comments 9:

Altogether, the quantitative analysis leads to confusing results which do not give a consistent picture of what might actually be happening in the experiments. Furthermore, the authors stress that the I-gel

resembles a “synthetic nucleus” that produces the siRNA. To strengthen this hypothesis, it would be necessary to have more information about the gel uptake and its fate inside the cell. This could be easily achieved with the fluorescently labeled X-DNA. An important question to answer is how stable the gels are inside the cell.

Our response: We thank the reviewer for the helpful comment. For the quantitative analysis, please refer the above responses.

The term “nucleus” has been used to simply reflect the function of nanoscale I-gel which is able to produce RNA in a cell. Therefore, we don’t intend to stress the I-gel as a “synthetic nucleus” and have removed the term from our revised manuscript.

In our original manuscript: pp2, 4, 17

This nanoscale hydrogel functioning like a nucleus will provide a novel platform technology for a highly efficient RNAi system.

The extraordinary high RNA productivity of Dgel and its gel structure inspired us to employ Dgel as a high performance RNA producing machinery that resembles a nucleus of which main function is synthesizing RNA within a cell.

We envision that the gel format of RNA producing will not only be applied for in vivo RNAi soon but also can be further developed for diverse RNA production such as mRNA in a cell and ultimately the gel will be employed as an in vivo RNA factory that is complementary to a nucleus.

In our revised manuscript: p2, 4, 17

This nanoscale hydrogel, which is able to produce RNA in a cell, provides a novel platform technology for a highly efficient RNAi system.

The extraordinarily high RNA productivity of the Dgel and its gel structure inspired us to employ it as a high-performance RNA-producing machinery within a cell.

We envision that the gel format of RNA production will not only be applied soon for in vivo RNAi, but will also be further developed for the production of diverse types of RNA, such as for mRNA in a cell.

Response to reviewer 2

Comments 1:

The authors have substantially revised the manuscript and satisfactorily responded to the comments. I have no further comments, except that the grammar still needs correcting throughout the manuscript.

Our response: We thank to the reviewer for the favorable comment. Following the reviewer's suggestion, we have taken a commercial correction service as certificated below to improve writing of our manuscript and help readers understand the results well.

CERTIFICATE OF ENGLISH EDITING					
This document certifies that the paper listed below has been edited to ensure that the language is clear and free of errors. The edit was performed by professional editors at Editage, a division of Cactus Communications. The intent of the author's message was not altered in any way during the editing process. The quality of the edit has been guaranteed, with the assumption that our suggested changes have been accepted and have not been further altered without the knowledge of our editors.					
TITLE OF THE PAPER A RNA producing DNA hydrogel as a platform for a high performance RNA interference system					
AUTHORS Jaejung Song ^{1,†} , Minhyuk Lee ^{2,†} , Taeyoung Kim ² , Jeongkyeong Na ³ , Yebin Jung ⁴ , Gyoo Yeol Jung ^{1,3} , Sungjee Kim ^{1,4} , Nokyoung Park ² ,					
JOB CODE MWHWA_1					
	Signature  Vikas Narang, Vice President, Author Services, Editage Date of Issue June 19, 2018				
Editage, a brand of Cactus Communications, offers professional English language editing and publication support services to authors engaged in over 500 areas of research. Through its community of experienced editors, which includes doctors, engineers, published scientists, and researchers with peer review experience, Editage has successfully helped authors get published in internationally reputed journals. Authors who work with Editage are guaranteed excellent language quality and timely delivery.	 CACTUS IDENTIFICATION SECURITY MANAGEMENT SYSTEM CERTIFIED 27001				
Contact Editage					
Worldwide request@editage.com +1 877-334-8243 www.editage.com	Japan submissions@editage.com +81 03-6958-3348 www.editage.jp	Korea submit@editage.com korea@editage.com 1544-8241 www.editage.co.kr	China fabiao@editage.cn 400-005-6055 www.editage.cn	Brazil contato@editage.com 0800-892-20-97 www.editage.com.br	Taiwan submit@jobs@editage.com 02 2657 0306 www.editage.com.tw

Response to reviewer 4

Comments 1:

A few things are still speculative - e.g. the authors indicate in their response that they expect the T7 polymerase to remain outside the I-gels when co-incubated and that transcription takes place on the surface of the I-gel only. However, the schematic in Figure 1 could be interpreted by readers as T7 polymerase being inside the I-gel (As Reviewer 3 did as well) - this needs to be clearly corrected and clarified. However, what is the evidence that the T7 polymerase (which is 99kd) is not entering the I-gel when co-incubated? Most hydrogels are extremely porous and allows fairly free movement of proteins. If they do go in, it could be an important distinction with Lipofectamine. Some experiments to address this is important - e.g. a simple nanogold labeling of the T7 followed by TEM of the I-gel could show its location. Or super-resolution microscopy could be employed. Regardless - the speculation that they are on the outside only is not supported by any data.

Our response: We thank the reviewer for the helpful comment. We have fully understood the reviewer's concern. The mention "the T7 RNA polymerases have not been encapsulated inside the I-gel and we expect that the polymerases exist out of the gel and also the polymerization mainly happened on the surface of the gel" that we had made in previous response was based on the size of the I-plasmid which has been crosslinked with X-DNA producing the nanoscale I-gel. In this experiment, we have fabricated I-gel of which size was around 100 nm in diameter and the number of base pair of the I-plasmid was around 5,000 whose length can reach easily up to more than 50 nm considering DNA's persistent length in aqueous buffer solution. Therefore, it has not been feasible to tell where the plasmid was located and, in this situation, we have thought that the discrimination of inside and surface of the I-gel might not be very meaningful. Since the schematic in Figure 1, however, still may induce confusion and too much out scaled, we have revised the Figure accordingly as shown in Fig. R1.

Figure R1. Illustration of RNAi mechanism of I-gel.

(a) A schematic diagram illustrating the I-gel mechanism in a living cell. (b) A schematic diagram illustrating the I-gel mechanism in a cell lysate assay.

Comments 2:

In response to reviewer comments the authors performed simple DNase digestion experiments. Although this had been classically employed by the gene delivery community, it's relevance is questionable. Where inside the cell are DNases and at this concentration? If the authors want to demonstrate stability, is there a way to extract the plasmid from the I-gel and run it in the gel and compare with the free plasmid. Most delivery formulations show stability in this manner. In fact a functional assay (transfection) with the extracted plasmid would be better. This hypothesis that the I-gel protects the plasmid better than naked I-plasmid inside the cell remains speculative.

Our response: We thank the reviewer for the helpful comment. The results in Fig. S14 in our original Supplementary information was obtained in a very harsh condition where the DNase concentration is much higher than natural conditions and shows the I-gel is stable even under the condition that the I-plasmid is completely decomposed. Considering the manufacturer's information that 1 unit of DNase I is around 0.2 ~ 0.5 ug in terms of

weight, the DNase concentration used in Fig. S14 was 666 ~ 1,665 pg/uL. This concentration is at least 83 times higher than that of kidney (~ 8 pg/uL). It has been known that kidney has highest DNase I concentration among mammalian cells except digestion related organs (Lacks, S.A. Deoxyribonuclease I in mammalian tissues. J. Biol. Chem. 256, 2644–2648 (1981).

Referring the reviewer’s comments, we carried out additional experiments. 1) I-gel and free I-plasmid were separately incubated in a DNase I solution of which concentration was set to be similar with that of kidney cell (6.66 ~ 16.65 pg/uL of Dnase I in the digestion solution). 2) After the digestion experiments, the I-gel and free I-plasmid has been used as templates for shRNA transcription by adding them in a transcription mixture separately followed by RT-qPCR for shRNA quantification.

1) As shown in the results (Fig. R2), the band intensities of I-gel of digestion have been changed little compared with that of time zero meaning that the I-gel remained in intact form even after 24 h of digestion while the band intensities of free I-plasmid have been significantly blurred.

2) According to the calculation based on the RT-qPCR results, the number of shRNA transcribed from each plasmid in I-gel and free I-plasmid showed big difference and the results were summarized in Table R1.

Although our new experiments were not performed in cells, the results seems to be enough to strongly support the I-gel’s stability over free I-plasmid and we have revised our manuscript accordingly.

Figure R2. Stability of I-plasmid and I-gel. Gel electrophoresis images of (a) I-gel and (b) I-plasmid treated with DNase I at 37 °C, lane 1: 0, lane 2: 1, lane 3: 4, lane 4: 12, lane 5: 24, lane 6: 48 hrs. (c) Time series of

shRNA amounts transcribed from I-gel and free I-plasmid. Error bars represent standard deviations from three replicates

Table R1. shRNAs transcription rate per plasmid template.

DNase I treated time	sample	Conc. of shRNA (nM)	Number of shRNA per plasmid template
0 h	I-plasmid	13.09 ± 0.29	18.71 ± 0.42
	I-gel	61.54 ± 3.15	87.91 ± 4.50
24 h	I-plasmid	4.82 ± 0.18	6.89 ± 0.25
	I-gel	65.39 ± 3.37	93.41 ± 4.81
48 h	I-plasmid	3.06 ± 0.05	4.37 ± 0.08
	I-gel	40.99 ± 3.38	58.55 ± 4.82

In our original manuscript: pp14, 18

The plasmid stability in a gel network has been confirmed by comparing the denaturation resistance of free I-plasmid and I-gel in a DNase existing solution. As shown in Supplementary Figure S14, the I-plasmid in the solution has almost completely digested in 10 minute of incubation. In case of I-gel, on the contrast, around 90% of the gel has remained even after 60 minute of incubation with DNase (measured using ImageJ).

I-gel stability test. Seven samples were prepared for each of the free I-plasmid and I-gel. In each sample, 2 µL of the free I-plasmid (57 ng) or I-gel (1:1,500 gel containing 57 ng of I-plasmid) was diluted in 12 µL of distilled water then 4 µL of transcription optimized 5× buffer was added and treated with 3.33×10^{-5} units of DNase I (No. M6101; Promega) to 20 µL of final volume, followed by incubation at 37 °C for 0, 1, 4, 12, 24, and 48 h, respectively. After incubation, the 20 µL samples were separated into two 10 µL aliquots, one for electrophoresis and another for following transcription. In each of the aliquot, the denaturation reaction was terminated by the addition of 1 µL of RQ1 DNase stop solution and incubation for 10 min at 65 °C. Thereafter, 10 µL of each sample was mixed with 2 µL of gel loading buffer and electrophoresed on a 2% agarose gel at 100 V for 60 min. Another aliquot of each sample was used for transcription reaction to demonstrate the transcription efficiency of I-gel after digestion reaction. shRNAs were transcribed from the I-gel or free I-plasmid templates using transcription kit (Riboprobe; Promega) following the manufacturer’s instruction. The obtained shRNAs were quantified by RT-qPCR as described in ‘Total RNA preparation and reverse transcription’ and ‘Real-time PCR and data analysis’ sections.

In our revised manuscript: pp14, 18

The plasmid stability in the gel network was confirmed by comparing the denaturation resistance of free I-plasmid and I-gel in a DNase-containing solution and followed transcription efficiency measurement of I-gel and free I-plasmid after digestion reaction. As shown in Supplementary Figure S13 and Table S3, the I-plasmids in the

solution were significantly denatured remaining only 40% of the free I-plasmid in 24 h of incubation losing about 63% of transcription efficiency. In contrast, approximately 95% of the I-gel had remained even after 24 h of incubation with DNase (measured using ImageJ software) keeping similar transcription efficiency with that of non-digested I-gel.

I-gel stability test. Seven samples were prepared for each of the free I-plasmid and I-gel. In each sample, 2 μ L of the free I-plasmid (57 ng) or I-gel (1:1,500 gel containing 57 ng of I-plasmid) was diluted in 12 μ L of distilled water then 4 μ L of transcription optimized 5 \times buffer was added and treated with 3.33×10^{-5} units of DNase I (No. M6101; Promega) to 20 μ L of final volume, followed by incubation at 37 $^{\circ}$ C for 0, 1, 4, 12, 24, and 48 h, respectively. After incubation, the 20 μ L samples were separated into two 10 μ L aliquots, one for electrophoresis and another for following transcription. In each of the aliquot, the denaturation reaction was terminated by the addition of 1 μ L of RQ1 DNase stop solution and incubation for 10 min at 65 $^{\circ}$ C. Thereafter, 10 μ L of each sample was mixed with 2 μ L of gel loading buffer and electrophoresed on a 2% agarose gel at 100 V for 60 min. Another aliquot of each sample was used for transcription reaction to demonstrate the transcription efficiency of I-gel after digestion reaction. shRNAs were transcribed from the I-gel or free I-plasmid templates using transcription kit (Riboprobe; Promega) following the manufacturer's instruction. The obtained shRNAs were quantified by RT-qPCR as described in 'Total RNA preparation and reverse transcription' and 'Real-time PCR and data analysis' sections.

Comments 3:

A key aspect of the authors response is the comparison with Lipofectamine. Supplementary Fig S9 showed that although similar amount of I-plasmid went into the cells whether it was delivered by I-gel or Lipofectamine. Yet - the GFP expression was much better silenced by I-gel than Lipofectamine. The authors speculated that this because I-gel allowed better transcription than lipofectamine. However, the variance in figure 4e makes it difficult to support this conclusion. At the minimum Fig 4 should show statistical significance in all groups through appropriate analyses. Also a larger number of "n" may mitigate the variance problem seen in Lipofectamine+I-plasmid mediated shRNA production and allow for a cleaner conclusion.

Our response: We thank the reviewer for the helpful comment. Following the reviewer's suggestion, we performed statistical analysis through the results. T-test was carried out on the data of Fig. 2c, Fig. 3b, c, e, f, and Fig. 4d, e and p-value of each test was noted on each data as shown in Figs. R3, R4 and R5. In addition, we have doubled the "n" of Fig. 4e from three to six by performing additional experiments. Although the statistical significance has not been significantly affected with the increased number of "n" in this case (see Fig. R5e), we have substituted the original data with the newly obtained one.

Figure R3. Optimization and evaluation of I-gel. (a) The FL spectra of GFP in each condition (no plasmid, only GFP expression, addition of scrambled shRNA 1eq, scrambled shRNA 100eq, scrambled plasmid mixture, scrambled plasmid gel, I-plasmid, and I-gel). (b) A comparison of the fluorescence intensities at the peak points of the graph (a) and the inset is fluorescence images of each sample. The relative FL intensity (c) with the addition of increasing amount of I-plasmid and I-gel ($*P < 0.01$, $**P < 0.001$ and $***P < 0.0001$ are noted between I-plasmid and I-gel), (d) with the variation of X-DNA and I-plasmid ratio in I-gel. For all experiments, 57 ng of I-plasmid has been used except (c). This data was obtained by *in vitro* transcription/translation with HeLa cell lysate system. Error bars represent standard deviations from three replicates.

Figure R4. Comparison of RNA expressions. (a, d) The RT-PCR products in agarose gel electrophoresis for (a) shRNA and (d) GFP mRNA detection in extracted RNA samples from the cell lysates. (a) Samples on gel are lane M: Molecular weight standard marker; **cDNAs reverse transcribed respectively from** Lane 1: standard shRNA, Lane 2: shRNA expressed from I-plasmid, Lane 3: shRNA expressed from I-gel; (d) **cDNAs reverse transcribed respectively from the GFP mRNAs produced in the cell lysates transcribed/translated in the presence of** Lane 1: only GFP plasmid, Lane 2, 3, 4, 5, 6: the addition of scrambled shRNA, scrambled plasmid mixture, scrambled plasmid gel, I-plasmid, and I-gel. (b, e) Copy number of RNA expressed from a template plasmid for (b) shRNA and (e) GFP mRNA expression in each condition (* $P < 0.01$, ** $P < 0.001$ and *** $P < 0.0001$ are noted between condition marked by line and I-gel). (c, f) The relative RNA expression value for (c) shRNA and (f) GFP mRNA in 5.7 ng, 57 ng and 114 ng of I-plasmid amount condition. The insets of (c) and (f) represent zoomed up images of dashed box (* $P < 0.01$, ** $P < 0.001$ and *** $P < 0.0001$ are noted between I-plasmid and I-gel). Error bars represent standard deviations from three replicates.

Figure R5. Gene-silencing effect by I-gel in a cellular level. (a) Comparison of intracellular uptake of plasmids in different formats. Second from left: I-plasmid labeled with Cy5, third: I-plasmid labeled with Cy5 complexed with lipofectamine, and the last: I-gel labeled with Cy5. Scale bar: 20 μm . (b) Fluorescence images of MDCK GFP-expressing (MDCK-GFP) cells co-incubated with I-plasmid, I-plasmid complexed with lipofectamine, and I-gel. Scale bar: 20 μm . (c) Fluorescence-activated cell sorter analysis of GFP-expressing cell line, MDCK-GFP cells after treatment of sample/polymerase complexes in serum-deficient medium. The number means the percentage of GFP-overexpressing cells sorted within a pre-fixed gate region as indicated by a bar. (d, e) The relative amounts of (d) GFP mRNA and (e) shRNA from GFP expressing MDCK cells after incubation in various conditions. ([#]used RNA amounts were 100-fold to the number of plasmid considering the RNA transcription rate of the I-gel, *P < 0.01, **P < 0.001 and ***P < 0.0001 are noted between marked condition and (c) only cell or (d) I-gel). Error bars represent standard deviations from three (d) or six (e) replicates.

Please address all correspondence to:

Nokyoung Park

Department of Chemistry

Myongji University, Yongin, South Korea

pospnk@mju.ac.kr

Thank you again for considering our work for publication, and please do not hesitate to contact us if you have any questions or require additional information.

Sincerely,

Authors

Reviewers' comments:

Reviewer #1 (Remarks to the Author):

The paper was substantially revised and should be accepted for publication

Reviewer #4 (Remarks to the Author):

The authors responded to the 3 comments from Rev 4. Although the response to comment 2 is adequate there are still gaps on the responses to comment 1 and comment 3. On Comment 1 - it remains unclear as to what the structure of the i-gel really is and whether the schematic shown is still correct or not. I would still suggest some insight on this is important for a high impact journal like Nature Communications. A TEM of the I-Gel could be informative.

On comment 3 - the authors indicate they increased "n", but from the figure legend it appears that was only for Figure 5(e) and Not for Figure 5(d)? IS that correct. The silencing data in Figure 5(d) doesn't show statistical difference between I-Gel and Lipo+I-plasmid, is that interpretation accurate? If so - then the impact of the I-gel system is lessened.

Also - most importantly, the authors mentioned they performed t-test on all the data to show significance - that is not adequate. For multiple group comparison Annova needs to be performed followed by appropriate corrections as needed. The statistical analyses needs to be corrected.

Response to Reviewers:

Manuscript ID: NCOMMS-17-24306C

Title: A RNA producing DNA hydrogel as a platform for a high performance RNA interference system

Response to reviewer 1

Comments 1:

The paper was substantially revised and should be accepted for publication

Our response: Our response: We thank to the reviewer for the favorable comment.

Response to reviewer 4

Comments 1:

The authors responded to the 3 comments from Rev 4. Although the response to comment 2 is adequate there are still gaps on the responses to comment 1 and comment 3. On Comment 1 - it remains unclear as to what the structure of the i-gel really is and whether the schematic shown is still correct or not. I would still suggest some insight on this is important for a high impact journal like Nature Communications. A TEM of the I-Gel could be informative.

Our response: We thank the reviewer for the helpful comment. As the reviewer pointed out the structure of the I-gel is not perfectly clear. Following the reviewer's suggestion, we have performed super resolution SEM imaging on the I-gel (In case of TEM, we could not observe the nanostructure of the I-gel. It might due to the three dimensional structure of the gel). The super resolution SEM image was added to Supplementary Fig. S19 in our revised Supplementary Information (Fig. S19d). As shown in the images of Fig. S19, the I-gel is composed of many micro- and nano-pores of which sizes are sometimes large enough and sometimes too small for T7 polymerase (around 5-10 nm) to enter into the inside of the gel. This image can help the readers to understand the structure of the I-gel. Although the exact location of the I-plasmid or knowing if the T7 need to enter into the inside of the gel to transcribe the plasmid may need more detailed study, we still would keep the schematic for this paper.

Supplementary Figure S19. SEM images of I-gel. (a, b) SEM images of large scale I-gel and (c, d) nano-scale I-gel at the ratio of 1,500:1 of X-DNA and plasmid.

Comments 2:

On comment 3 - the authors indicate they increased "n", but from the figure legend it appears that was only for Figure 5(e) and Not for Figure 5(d)? IS that correct. The silencing data in Figure 5(d) doesn't show statistical difference between I-Gel and Lipo+I-plasmid, is that interpretation accurate? If so - then the impact of the I-gel system is lessened.

Our response: We thank the reviewer for the helpful comment. We have increased 'n' of the experiment in Fig. 4e, where the reviewer pointed out that at previous revision for clearer conclusion. The results of Fig. 4d have relatively low standard deviations, so we focused more on Fig. 4e experiments and interpretation of the results. The reviewer have pointed out that the statistical difference of GFP mRNA is inconsiderable between Lipo+I-plasmid and I-gel. In Fig. 4d, the difference of GFP mRNA and p value between Lipo+I-plasmid and I-gel condition was 2.64-fold and 0.0133, respectively. When comparing the effect of the two conditions, this shows a considerable difference in the target RNA expression and means that the probability that the present comparison of results is a null hypothesis is quite low. This indicates a statistically significant difference. Also, the main topic of this paper is the novel approach of the effective and convenient RNAi methods. Although we have studied that other positive conditions can have similar effect to the I-gel for scientific validation, in limited pages, it is reasonable to compare and analyze the effects taking into account the various new points of I-gel. For an accurate comparison between I-gel and I-plasmid, not only Fig. 4d but also Fig. 4e should be taken into consideration. If the plasmid is used for intended effect with insertion into the cell, it should be supplemented with delivery reagent such as Lipofectamine. As a result of our experiments, I-gel can enter into cells without any supporting delivery reagent, unlike I-plasmid, and the stability against DNase is also higher than that of I-plasmid. Also, most importantly, the shRNA production of I-gel was about 9 times higher than that of Lipo+I-plasmid in cellular condition. Although there is no perfect 1:1 correlation between shRNA and mRNA levels at the cellular level, the amount of shRNA produced is clearly different. As a result, the difference in RNAi effect can be attributed to the 9-fold shRNA expression difference. We believe that this indicates that the I-gel system is obviously meaningful.

Following the reviewer's suggestion, we performed additional statistical analysis through the results and revised our manuscript accordingly.

Figure 2. Optimization and evaluation of I-gel. (a) The FL spectra of GFP in each condition (no plasmid, only GFP expression, addition of scrambled shRNA 1eq, scrambled shRNA 100eq, scrambled plasmid mixture, scrambled plasmid gel, I-plasmid, and I-gel). (b) A comparison of the fluorescence intensities at the peak points of the graph (a) and the inset is fluorescence images of each sample. The relative FL intensity (c) with the addition of increasing amount of I-plasmid and I-gel ($*P < 0.05$ and $***P < 0.001$ are noted between I-plasmid and I-gel), (d) with the variation of X-DNA and I-plasmid ratio in I-gel. For all experiments, 57 ng of I-plasmid has been used except (c). This data was obtained by in vitro transcription/translation with HeLa cell lysate system. Error bars represent standard deviations from three replicates.

Figure 3. Comparison of RNA expressions. (a, d) The RT-PCR products in agarose gel electrophoresis for (a) shRNA and (d) GFP mRNA detection in extracted RNA samples from the cell lysates. (a) Samples on gel are lane M: Molecular weight standard marker; cDNAs reverse transcribed respectively from Lane 1: standard shRNA, Lane 2: shRNA expressed from I-plasmid, Lane 3: shRNA expressed from I-gel; (d) cDNAs reverse transcribed respectively from the GFP mRNAs produced in the cell lysates transcribed/translated in the presence of Lane 1: only GFP plasmid, Lane 2, 3, 4, 5, 6: the addition of scrambled shRNA, scrambled plasmid mixture, scrambled plasmid gel, I-plasmid, and I-gel. (b, e) Copy number of RNA expressed from a template plasmid for (b) shRNA and (e) GFP mRNA expression in each condition (\uparrow concentration of scrambled shRNA was 102 fold increased in consideration of the template to RNA transcription rate of the I-gel) (b) (** $P < 0.001$ is noted between I-plasmid and I-gel) (e) (** $P < 0.001$ is noted between each condition marked by line and I-gel). (c, f) The relative RNA expression value for (c) shRNA and (f) GFP mRNA in 5.7 ng, 57 ng and 114 ng of I-plasmid amount condition. The insets of (c) and (f) represent zoomed up images of dashed box (** $P < 0.001$ is noted between I-plasmid and I-gel). Error bars represent standard deviations from three replicates.

Figure 4. Gene-silencing effect by I-gel in a cellular level. (a) Comparison of intracellular uptake of plasmids in different formats. Second from left: I-plasmid labeled with Cy5, third: I-plasmid labeled with Cy5 complexed with lipofectamine, and the last: I-gel labeled with Cy5. Scale bar: 20 μ m. (b) Fluorescence images of MDCK GFP-expressing (MDCK-GFP) cells co-incubated with I-plasmid, I-plasmid complexed with lipofectamine, and I-gel. Scale bar: 20 μ m. (c) Fluorescence-activated cell sorter analysis of GFP-expressing cell line, MDCK-GFP cells after treatment of sample/polymerase complexes in serum-deficient medium. The number means the percentage of GFP-overexpressing cells sorted within a pre-fixed gate region as indicated by a bar. (d, e) The relative amounts of (d) GFP mRNA and (e) shRNA from GFP expressing MDCK cells after incubation in various conditions. ([†]concentration of these conditions were 102 fold increased in consideration of the template to RNA transcription rate of the I-gel) (d: *P < 0.05 is noted between each condition marked by line and Lipofectamine-shRNA (or I-gel), #P < 0.05 is noted between each condition and shRNA) (e: *P < 0.05 and **P < 0.005 are noted between each condition and I-gel). Error bars represent standard deviations from three (d) or six (e) replicates.

Comments 3:

Also - most importantly, the authors mentioned they performed t-test on all the data to show significance - that is not adequate. For multiple group comparison Annova needs to be performed followed by appropriate corrections as needed. The statistical analyses needs to be corrected.

Our response: We thank the reviewer for the helpful comment. Although we think the T-test is still an appropriate statistical approach to compare the efficiency of I-gel with other controls, following the reviewer's suggestion and since we also recognize that repeating T-test for multiple groups may decrease reliability of the test, we have additionally performed F-test for the data in Fig. 4d. As summarized in Supplementary Table S3, the GFP mRNA inhibition effect of 1st positive control including Lipo+I-plasmid and 2nd positive control including I-gel showed statistically significant difference. Thus, the additional F-test also supports our conclusion that the I-gel was more efficient in the interference performance than that of other controls including Lipo+I-plasmid. The table has been added to the revised Supplementary Information and we have revised our manuscript accordingly.

In our revised manuscript: p13

To confirm the significance of the GFP mRNA inhibition efficiency difference between the controls and I-gel, we have additionally performed F-test among the multiple groups of controls and I-gel (Supplementary Table S3).

Supplementary Table S3. F test result of GFP mRNA level for multiple group analysis among the controls and I-gel.

Group	Degree of freedom (btw. , w/i)	F ratio	P value	F critical value (P<0.05)	F critical value (P<0.01)
Negative control (NC)	4, 10	0.7761	0.5653	3.48	5.99
1st Positive Control (PC)	2, 6	1.4828	0.2997	5.14	10.92
NC : 1st PC	7, 16	2.9274	0.0356*	2.66	4.03
NC : 2nd PC	6, 14	8.9194	0.0004**	2.85	4.46
1st PC : 2nd PC	4, 10	6.3119	0.0084**	3.48	5.99

Each group includes the following conditions; Negative control: Cell only, scrambled shRNA, scrambled plasmid mixture, Lipo + scrambled plasmid mixture and scrambled plasmid gel. 1st positive control: I-plasmid, Lipo + I-plasmid and shRNA. 2nd positive control: Lipo + shRNA and I-gel. The p value within *P < 0.05 and **P < 0.01 level cannot be valid for null hypothesis, respectively. Non-marked group cannot reject null hypothesis because p > 0.05.

REVIEWERS' COMMENTS:

Reviewer #4 (Remarks to the Author):

Most of the concerns of the reviewer is addressed - however, the statistics is still unclear. The authors performed an F test but doe snot explain why, whether the data was normal or not, etc. Given the authors used bar graphs and not box plots or scattered plots it is difficult for the reviewer to know what the distribution of the data looks like. There should actually be a statistics section that explains the type of distribution the data has and why a particular statistics was chosen.

Also, in response to Comment 2 the authors indicate a fold change and a p value - what statistics was used for that? Please indicate if that was a multiple comparison test. The conclusions being drawn on the mRNA data is dependent on statistics.

Response to Reviewers:

Manuscript ID: NCOMMS-17-24306D

Title: A RNA producing DNA hydrogel as a platform for a high performance RNA interference system

Dear Editor,

Thank you so much for the correspondence with the referees. We appreciate all the sincere comments by reviewers on our manuscript. The comments were carefully considered during the revision of the manuscript. In the upcoming pages, we tried to respond to these comments one by one. Our detailed list of changes and responses to reviewers' concerns are given below.

Response to reviewer 4

Comments 1:

Most of the concerns of the reviewer is addressed - however, the statistics is still unclear. The authors performed an F test but do not explain why, whether the data was normal or not, etc. Given the authors used bar graphs and not box plots or scattered plots it is difficult for the reviewer to know what the distribution of the data looks like. There should actually be a statistics section that explains the type of distribution the data has and why a particular statistics was chosen.

Also, in response to Comment 2 the authors indicate a fold change and a p value - what statistics was used for that? Please indicate if that was a multiple comparison test. The conclusions being drawn on the mRNA data is dependent on statistics.

Our response: We thank the reviewer for the helpful comment. The authors agreed to the reviewer's opinion that it is easier to know the distribution of data with a scattered type graph than a bar type one. The authors, however, think that a bar type graph is also helpful to the readers in a view of comparing the produced mRNA amounts in a more immediate way. Therefore, we would like to present the mRNA quantification data in both two different types of graph, one in a bar type (Fig. 5c) and another in scattered type (Supplementary Figure 10) in our revised manuscript.

In addition, to confirm the distribution of data follows a normal distribution, the Shapiro-Wilk test was performed. From the result of Shapiro-Wilk test, our data were observed to be normally distributed. With previous comments from the reviewer, we have proceeded with one-way ANOVA instead of repeating T-tests for limiting the alpha inflation and more statistically accurate analysis. So, one-way ANOVA was performed for all experimental groups of Fig. 4d and then Holm-bonferroni post-test was performed for further statistical analysis. The proceeding the Holm-Bonferroni post-test have the advantage of reducing possibility of type I error and alpha inflation, and we believe that this will be useful for our data analysis (*Clin. Exp. Pharmacol. Physiol.* **1998**, 25, 1032-1037, *Am. J. Public Health* **1996**, 86, 726-728).

Briefly, alpha expansion, also referred to as the Type I error rate, indicates the possibility of false conclusion

when a series of hypothesis tests are performed. In a statistical hypothesis test, Type I errors are errors that wrongly reject the true null hypothesis, and Type II errors are errors that wrongly adopt the null hypothesis to be. In our case, we selected the Holm-Bonferroni post-test to reduce the type I error because the type I error is more fatal in our data. The type I error can lead to false conclusions that there is a difference between the experimental group and the control group despite no statistical difference. To perform the Holm-Bonferroni (HB) post-test, we have performed HB post-test with reference to other research article and the results of the Bonferroni post-test results obtained from Prism 7.05 software which is a statistical analysis program (*Scand J Statistics* **1979**, 6, 65-70).

The HB post-test was performed for I-gel and cell only condition, respectively. When the I-gel condition was compared with other groups, the I-plasmid ($P = 0.058$), Lipo+I-plasmid ($P = 0.175$), shRNA (102-fold) ($P = 0.347$) and Lipo+shRNA (102 fold) ($P = 0.983$) conditions were observed with no significant difference. When the I-gel condition was compared with other groups, considering the following results, the I-plasmid and Lipo+I-plasmid conditions did not show statistically significant differences with that, but P-value of HB post-test with I-plasmid were relatively low compared to other groups ($P = 0.058$). When the cell only condition was compared with other groups, only the Lipo+shRNA (102 fold) ($P = 0.012$) and I-gel ($P = 0.011$) conditions showed statistically significant differences. In conclusion, although there was no statistically significant difference between positive conditions (I-plasmid, Lipo+I-plasmid, shRNA (102 fold), Lipo+shRNA (102 fold), I-gel), it was statistically confirmed that I-gel and Lipo+shRNA (102 fold) conditions affected significant inhibition of GFP mRNA. Based on these statistical analyzes, the RNA interfering ability of I-gel is different when compared to I-plasmid or shRNA alone. We think that these differences of RNA interfering ability are the increased shRNA transcription rate by the I-gel system. For better understanding of readers and reviewers, we have added the detail procedure of statistical analysis in Method section.

(in Method section)

Statistical Analysis

Statistical analysis was performed with Prism 7.05 software (GraphPad Software) for Student's t-test, one-way ANOVA with Bonferroni multiple comparisons post-test. The Holm-Bonferroni multiple comparisons post-test was performed using Bonferroni multiple comparisons post-test results from Prism 7.05 software (GraphPad Software)³⁰. The normality test was calculated by the Shapiro-Wilk test. In results of the Shapiro-Wilk test, Data were approximately normally distributed. Statistical significance is indicated as *P < 0.05, **P < 0.01, ***P < 0.001.

Supplementary Figure 10. Scattered plot type of the relative GFP mRNA amounts.

(†concentration of these conditions were 102 fold increased in consideration of the template to RNA transcription rate of the I-gel, *P < 0.05, analyzed by one-way ANOVA, followed by Holm-Bonferroni multiple comparisons post-test. Asterisks indicate statistically significant differences between each condition and Cell only)

Supplementary Table 3. One-way ANOVA results of GFP mRNA level for multiple group analysis.
(The case of that the cell only condition was compared with other groups)

i) One way ANOVA table

ANOVA table	SS	DF	MS	F (DFn, DFd)	P value
Treatment (between columns)	3.1064	9	0.3452	F (9, 20) = 5.5477	P=0.0007
Residual (within columns)	1.2443	20	0.0622		
Total	4.3507	29			

Note: (SS: Sum of Squares, DF: Degree of Freedom, MS: Mean Square)

ii) Holm Bonferroni's multiple comparisons test table (Cell only vs)

Holm Bonferroni's multiple comparisons test	Mean Diff.	t	Holm P Value	Summary
Cell only vs. Scrambled shRNA	-0.1995	0.9797	>0.9999	ns
Cell only vs. Scrambled plasmid mixture	-0.1354	0.6647	>0.9999	ns
Cell only vs. Scrambled plasmid gel	0.0674	0.3308	>0.9999	ns
Cell only vs. Lipo+Scrambled plasmid mixture	0.1038	0.5096	>0.9999	ns
Cell only vs. I-plasmid	0.2085	1.024	>0.9999	ns
Cell only vs. Lipo+I-plasmid	0.3446	1.692	0.6368	ns
Cell only vs. shRNA	0.4663	2.290	0.2313	ns
Cell only vs. Lipo+shRNA	0.7582	3.723	0.0121	*
Cell only vs. I-gel	0.7537	3.701	0.0113	*

Note: (t: Bonferroni T-statistic)

* Asterisks indicate statistically significant differences between each group and Cell only.

iii) Holm Bonferroni's multiple comparisons test table (I-gel vs)

Holm Bonferroni's multiple comparisons test	Mean Diff.	t	Holm P Value	Summary
I-gel vs. Cell only	-0.7537	3.701	0.0099	*
I-gel vs. Scrambled shRNA	-0.9532	4.681	0.0013	**
I-gel vs. Scrambled plasmid mixture	-0.8891	4.365	0.0024	**
I-gel vs. Scrambled plasmid gel	-0.6863	3.370	0.0183	*
I-gel vs. Lipo+scrambled plasmid mixture	-0.6499	3.191	0.0229	*
I-gel vs. I-plasmid	-0.5452	2.677	0.0580	ns
I-gel vs. Lipo+I-plasmid	-0.4091	2.009	0.1748	ns
I-gel vs. shRNA	-0.2874	1.411	0.3472	ns
I-gel vs. Lipo+shRNA	0.0045	0.0219	0.9828	ns

Note: (t: Bonferroni T-statistic)

* Asterisks indicate statistically significant differences between each group and I-gel.

Please address all correspondence to:

Nokyoung Park

Department of Chemistry

Myongji University, Yongin, South Korea

pospnk@mju.ac.kr

Thank you again for considering our work for publication, and please do not hesitate to contact us if you have any questions or require additional information.

Sincerely,

Authors